# Cannabimimetic *N*-Stearoylethanolamine as “Double-Edged Sword” in Anticancer Chemotherapy: Proapoptotic Effect on Tumor Cells and Suppression of Tumor Growth versus Its Bio-Protective Actions in Complex with Polymeric Carrier on General Toxicity of Doxorubicin In Vivo

**DOI:** 10.3390/pharmaceutics15030835

**Published:** 2023-03-03

**Authors:** Rostyslav Panchuk, Nadiya Skorokhyd, Vira Chumak, Lilya Lehka, Halyna Kosiakova, Tetyana Horid’ko, Iehor Hudz, Nadiya Hula, Anna Riabtseva, Nataliya Mitina, Alexander Zaichenko, Petra Heffeter, Walter Berger, Rostyslav Stoika

**Affiliations:** 1Institute of Cell Biology National Academy of Sciences of Ukraine, Drahomanov Str., 14/16, 79005 Lviv, Ukraine; 2Center for Cancer Research and Comprehensive Cancer Center, Medical University of Vienna, Borschkegasse 8a, 1090 Vienna, Austria; 3Palladin Institute of Biochemistry National Academy of Sciences of Ukraine, Leontovycha Str. 9, 01030 Kyiv, Ukraine; 4Department of Organic Chemistry, Lviv Polytechnic National University, S. Bandera Str. 12, 79013 Lviv, Ukraine; 5Department of Applied Physics and Nanomaterial Science, Lviv Polytechnic National University, S. Bandera Str. 12, 79013 Lviv, Ukraine

**Keywords:** doxorubicin, *N*-stearoylethanolamine, polymeric carriers, tumor cells, drug resistance, apoptosis, mice, blood

## Abstract

This study reports a dose-dependent pro-apoptotic action of synthetic cannabimimetic *N*-stearoylethanolamine (NSE) on diverse cancer cell lines, including multidrug-resistant models. No antioxidant or cytoprotective effects of NSE were found when it was applied together with doxorubicin. A complex of NSE with the polymeric carrier poly(5-(tert-butylperoxy)-5-methyl-1-hexen-3-yn-co-glycidyl methacrylate)-graft-PEG was synthesized. Co-immobilization of NSE and doxorubicin on this carrier led to a 2-10-fold enhancement of the anticancer activity, particularly, against drug-resistant cells overexpressing ABCC1 and ABCB1. This effect might be caused by accelerated nuclear accumulation of doxorubicin in cancer cells, which led to the activation of the caspase cascade, revealed by Western blot analysis. The NSE-containing polymeric carrier was also able to significantly enhance the therapeutic activity of doxorubicin in mice with implanted NK/Ly lymphoma or L1210 leukemia, leading to the complete eradication of these malignancies. Simultaneously, loading to the carrier prevented doxorubicin-induced elevation of AST and ALT as well as leukopenia in healthy Balb/c mice. Thus, a unique bi-functionality of the novel pharmaceutical formulation of NSE was revealed. It enhanced doxorubicin-induced apoptosis in cancer cells in vitro and promoted its anticancer activity against lymphoma and leukemia models in vivo. Simultaneously, it was very well tolerated preventing frequently observed doxorubicin-associated adverse effects.

## 1. Introduction

The low selectivity of action of anticancer drugs and their numerous negative side effects on normal tissues of cancer patients belong to the most significant drawbacks in chemotherapy [1,2]. Another reason for the low efficiency of cancer treatment is the development of acquired resistance of tumor cells to chemotherapeutic agents, which is frequently caused by the overexpression of specific drug-transporting proteins—ABCB1 (P-glycoprotein), ABCC1 (MRP-1), or ABCG2 (Bcrp) [3].

In order to reduce the general toxicity of anticancer drugs, novel approaches were developed for anticancer chemotherapy selectively targeting certain molecules or metabolic pathways overexpressed or hyperactivated in cancer cells [3]. However, for most patients, the high costs needed for such therapy are unacceptable. In addition, the application of targeted chemotherapy does not guarantee long-term remission, even when used in combination with traditional medicines.

Immobilization of highly toxic anticancer drugs on bio-functionalized delivery platforms can reduce the effective drug concentration as well as increase tumor specificity [4]. Here especially, nanocarriers are of interest, as they usually have a longer duration of action, increased bioavailability, and also reduced side effects in patients [5]. In addition, most modern drug delivery systems possess desirable characteristics as a possibility for directed delivery of the drug to the location of the pathological process which is absent in the traditional forms of anticancer drugs.

The use of nanoparticles as highly effective, non-toxic, and biodegradable platforms for drug delivery to target cells has become widespread. For this purpose, liposomes, carbon, silicon, iron, and gold nanomaterials were proposed. However, they all have certain disadvantages preventing their broad use in clinical practice [6]. Here, we applied a new polymeric carrier based on poly(5-(tert-butylperoxy)-5-methyl-1-hexen-3-yn-co-glycidyl methacrylate)-graft-PEG for drug delivery. Previously, we showed that immobilization of the antitumor antibiotic doxorubicin (Dx) on similar polymeric oligoelectrolytes significantly enhanced the cytotoxic effect towards tumor cells [7].

Immobilization of the Dx on the polymeric carrier reduced its effective dose, however, it did not influence its general toxicity [7]. Therefore, in order to further decrease the general toxicity of the Dx in the body, it was proposed to functionalize the Dx-containing polymeric carrier with an additional ligand that is known to possess antioxidant and anti-inflammatory activity. To reach this aim, *N*-acylethanolamines (NAE) were chosen. These biologically active lipids belong to a group of cannabinoid-like compounds, that are produced by neurons and some other cells of the body in response to diverse physiological agents or stress [8].

Previously, we have established the neuroprotective effect of the synthetic NAE at chronic morphine addiction as well as their cardioprotective effect in an in vivo model of experimental heart ischemia [9,10]. Moreover, *N*-stearoylethanolamine (NSE), a representative of the NAE group, inhibited the growth of Lewis lung carcinoma in mice and reduced the extent of metastatic lung lesions in tumor-bearing animals [11]. Although the NAEs demonstrated cytostatic effects toward cancer cell lines, the underlying molecular mechanisms remained unclear. Therefore, in this study, the synthetic cannabimimetic NSE was chosen as a ligand for functionalization of the Dx-containing polymeric composite in order to improve the anticancer activity and decrease the severe adverse effects of Dx. A unique bi-functionality of the NSE (anticancer versus bioprotective effects) is demonstrated and potential mechanisms of such a dual role of this cannabimimetic are considered.

## 2. Materials and Methods

### 2.1. Chemical Part

#### Synthesis of the Polymeric Carrier (PC)

**Materials**. The peroxide monomer 5-tert-butylperoxy-5-methyl-1-hexen-3-yne (VEP) was synthesized, as described [12]. After vacuum distillation its characteristics were: [O] = 8.7%, d_4_^20^ = 0.867 g/mL, n_d_^20^= 1.4482. Glycidyl methacrylate (GMA) (Merck, Darmstadt, Germany) was purified by distillation in vacuum. The poly(ethylene glycol) methyl ether (mPEG, M_n_ = 750 Da), azobis(isobutyronitrile) (AIBN) and boron trifluoride ethyl etherate were obtained from Merck (Darmstadt, Germany) and used without additional purification. The solvents ethyl acetate, dioxane, chloroform (Chl), and dimethyl sulfoxide (DMSO) were purchased from Sigma-Aldrich and used after distillation.

**Synthesis**. The comb-like copolymer poly(VEP-co-GMA)-graft-mPEG was synthesized in the following order [13]. Initially, the polymer containing side epoxide groups was obtained via radical copolymerization of VEP (0.41 g, 2.2 mmol) and GMA (7.72 g, 54.4 mmol) in dioxane (7.9 mL), and AIBN (0.129 g, 0.8 mmol) was used as initiator. The polymerization was carried out at 343 K until 65% of monomer conversion was reached. In order to remove the catalyst (boron trifluoride⋅diethyl ether, BF_3_⋅OEt_2_), the obtained polymer was purified by re-precipitation from solution in dioxane into hexane [14]. Then, the copolymer was used as backbone for adjunction of mono-substituted mPEG molecules via reactions with side epoxide groups of GMA links. Boron (0.27 mL, 0.19 mmol) catalyst and 15 mL of the mPEG solution (2.5 g, 3.35 mmol) in dioxane were subsequently added in drops to 20 mL of the poly(VEP-co-GMA) solution (1.0 g) in dioxane and stirred for 3 h at 313 K. Residual mPEG was removed by the dialysis using dialysis bags with pore sizes MWCO 6–8 kDa (Merck, Darmstadt, Germany). The resulting comb-like polymer was dried until a constant weight at room temperature under vacuum was reached.

The composition of poly(VEP-co-GMA)-graft-mPEG was determined using elemental [15] and functional analysis [16]. The structure of polymer was also confirmed by NMR spectroscopy. The NMR spectra were recorded in the DMSO-d6 on a Bruker AV300 NMR spectrometer operating at a frequency of 300 MHz (Billerica, MA, USA). Number average (M_n_) and weight average (M_w_) molecular weights were determined using gel-penetration chromatography on a Styragel HR1 (THF) column (Waters GPC/HPLC Instrument, Waters, Milford, MA, USA). Tetrahydrofuran (THF) was used as eluent and polystyrene of known molecular weight were used as standard. The flow rate was 0.5 mL/min. The assumed structure of PEG-containing comb-like polymer is presented in the scheme (Figure 1).


**
*Preparing the waterborne system without Dx and NSE*
**


To prepare the PC-based micelles in aqueous solutions, 0.09 g of PC was dissolved in 0.9 mL DMSO. This PC solution was added to 8.0 mL 0.9% NaCl. Then, the solution was stirred for 0.5 h and sonicated for 10 s.


**
*Preparing the waterborne system without Dx but with NSE*
**


To prepare the PC- and NSE-based micelles in aqueous solutions, 0.09 g of PC was dissolved in a 0.45 mL mixture of DMSO: Chl (the ratio of DMSO: Chl = 3:1 mass), 0.03 g of the NSE was also dissolved in a 0.45 mL mixture of DMSO: Chl. PC and NSE solutions were mixed. This solution of PC and NSE was added to 8.0 mL 0.9% NaCl. Then, the solution was stirred for 0.5 h and sonicated for 10 s. The solution was stirred while heating at 40 °C for another 1 h to remove traces of Chl.


**
*Preparing the waterborne delivery system with Dx but without NSE*
**


Water dispersions of the micelle of PC with Dx were obtained by dropping the solution of Dx and PC in DMSO into water, in the following order: 0.09 g of PC was dissolved in 0.25 mL of DMSO and 0.003 g of Dx was dissolved in 0.20 mL of DMSO. The solutions of PC and Dx were mixed, added dropwise to 8.0 mL 0.9% NaCl and sonicated for 10 s.


**
*Preparing the waterborne delivery system with Dx and NSE*
**


The water dispersion of the PC- and NSE-based micelle complex loaded with Dx was obtained by dropping the solution of Dx, PC, and NSE into water, in the following order. First, 0.09 g PC, 0.003 g NSE, and 0.003 g Dx were dissolved in 0.2 mL, 0.15 mL, and 0.10 mL of a DMSO: Chl mixture, respectively. Then, the solutions were mixed, added to 8.0 mL 0.9% NaCl and sonicated for 10 s. Subsequently, the solution was stirred, while heating at 40 °C for another 1 h to remove traces of Chl. Subsequently, the aqueous dispersion was again treated by ultrasound for 10 s.

The sizes of the polymer micelles of PC and of the complexes of PC with Dx and NSE were measured by dynamic light scattering (DLS) using a Zetasizer Nano ZS (Malvern Instruments GmbH, Stuttgart, Germany) and DynaPro NanoStar (Wyatt Technology, Santa Barbara, CA, USA) and by photon correlation spectra using the NIBS (non-invasive back scatter) at 25 °C. The samples for DLS measurements were prepared as described above, and if necessary were diluted with bi-distilled water, pH 6.5–7.0. Three–five measurements were made for every sample (each measurement consisted of five cycles, the range between measurements was 5 min).

Zeta potential measurements were carried out with a Zetasizer Nano ZS at a fixed temperature of 25 °C. In total six measurements were performed at each water dispersion and the average value was calculated. The zeta potential data were reproducible within a precision of better than 5%. Table 1 presents some colloid chemical characteristics of aqueous dispersions. Figure 2 shows the general scheme for the formation of a mixed micellar structure.


**
*Studies of Dx release from drug delivery system*
**


Dx release from micellar complexes was studied in model experiment with saline using Pur-A-Lyzer™ Maxi Dialysis Kit (Merck, Darmstadt, Germany). An amount of 3 mL of solution of pure Dx or micellar complexes Dx-PC and Dx-PC-NSE were placed into Maxi dialysis tubes (pore size 3.5 kDa MWCO) and plunged in 30 mL of saline (0.9% sodium chloride). All flasks were incubated at 37 °C under mild agitation, and samples were collected at different time intervals. Release of Dx from the micellar complexes was analyzed by measuring UV spectra of dialysate at 220 nm on Nanodrop ND-1000 (Thermo Fisher Scientific, Waltham, MA, USA), and plotted as a function of time to determine drug release kinetics. All experiments were performed at least three times with good reproducibility.

### 2.2. Biological Part

#### 2.2.1. Materials

Dx hydrochloride was obtained from Pfizer (New York, NY, USA). NSE was synthesized at the Department of Lipids Biochemistry, Palladin Institute of Biochemistry (Kyiv, Ukraine) using the condensation of ethanolamine with stearic acid [17]. Purity of NSE was confirmed by gas chromatography (HRGC 5300, Carlo Erba Instruments, Val De Reuil, France) using packed column Chromosorb W 100–125 (Supelco, Montclair, NJ, USA) with phase 10% Silar 5 CP. The temperature of the injector and detector were 250 °C and 270 °C, respectively.

The sample of NSE for gas chromatography was prepared as follows: 0.1 mg of NSE, 1 mL of benzene, and 1 mL of 3 M HCl in methanol were mixed in a glass ampoule. The ampoule was sealed and heated in a water bath at 100 °C for 1.5 h. After this procedure, ampoule was cooled and opened. The solution was removed, filled into a sample tube, and mixed with 5 mL of hexane. The upper fraction of the solution was collected and transferred into a glass flask. The glass flask with the NSE sample was then dried up using a rotary evaporator at 35 °C. The dried sample was dissolved into 0.2 mL benzene. The solution was separated by a thin layer chromatography plate using benzene. The methyl ester of stearic acid was used as the marker probe. The zone, which consisted of methyl ester NSE, was extracted into benzene. Dried NSE powder was resuspended in ethanol to prepare 20 mM stock solutions. Serial dilutions of NSE were prepared in cell culture medium ex tempore before adding to cell culture.

#### 2.2.2. Cell Culture and Cytotoxicity Measurements

The human lung cancer cell line SW1573 and its ABCB1-overexpressing subline 2R160 were generously donated by H. Broxterman (Free University Hospital, Amsterdam, The Netherlands), human acute promyelocytic HL-60 leukemia cells and its ABCC1-overexpressing subline HL-60/adr were generously donated by M. Center (Kansas State University, KS, USA). The human breast adenocarcinomas MCF-7 and MDA-MB-231, the human T-leukemia Jurkat, the murine leukemia L1210, the human cervix carcinoma HeLa and the pseudonormal human HEK293 kidney cells were purchased from the American Type Culture Collection (Manassas, VA, USA). The murine NK/Ly lymphoma cells were obtained from the Kavetsky Institute of Experimental Oncology, Pathology, and Radiobiology, National Academy of Sciences of Ukraine (Kyiv, Ukraine).

SW1573 and SW1573/2R160 cells were cultured in DMEM medium (Merck, Darmstadt, Germany) supplemented with 10% (*v*/*v*) heat-inactivated fetal bovine serum (FBS, Merck, Darmstadt, Germany) and 50 μg/mL gentamicin (Merck, Darmstadt, Germany) in a humidified atmosphere containing 5% CO_2_ at 37 °C. All other cell lines were cultured in FCS- and gentamycin-supplemented RPMI-1640 medium. Cells were subcultured every 72 h at the rate of 5 × 10^5^ cells per 1 mL of culture medium for suspension cultures and 5 × 10^4^ cells per 1 mL of culture medium for substrate-dependent cultures. For morphologic assessment, cells were checked under an inverted microscope IB-100 (Delta Optical, Gdansk, Poland).

For cytotoxicity studies, cells were seeded into 24-well tissue culture plates (Greiner Bio-one, Frickenhausen, Germany) at a concentration of 5 × 10^5^ per well (suspension cell lines) and 10^5^ per well (adherent cell lines). Short-term (24 h) cytotoxic effects were studied under an Evolution 300 Trino microscope (Delta Optical, Gdansk, Poland) after cell staining with trypan blue dye (0.1%).

#### 2.2.3. Apoptosis Ana Lysis

For cell death analyses, cells were stained with APC-conjugated Annexin V and propidium iodide (PI) using an apoptosis detection kit (BD Biosciences, San Jose, CA, USA), according to the manufacturer’s instructions. In particular, 24 h after the addition of various concentrations of NSE and Dx, Jurkat T-leukemia cells were centrifuged at 1200 rpm, washed twice with PBS, and incubated for 15 min in Annexin V binding buffer (BD Biosciences, Franklin Lakes, NJ, USA) containing 1/50 volume of APC-conjugated Annexin V solution and PI (50 µg/mL). Then, samples were diluted 2 times by appropriate volume of Annexin V binding buffer (BD Pharmingen, USA) and immediately measured using 670/14 nm (APC) and 575/26 nm (PI) filters of BD LSR Fortessa flow cytometer (BD Biosciences, Franklin Lakes, NJ, USA). Analysis of the obtained results was carried out using Cell Quest Pro software (BD Biosciences, Franklin Lakes, NJ, USA).

#### 2.2.4. Cell Cycle Analysis

Impact of drug treatment on the cell cycle distribution of Jurkat cells was assessed according to protocol described by Walker et al. [18]. After drug treatment, 2 × 10^6^ cells were collected, pelleted by spinning at 1000 rpm, 4 °C for 5 min, resuspended in 1 mL of cold PBS, and fixed by adding drop-by-drop 4 mL of −20 °C absolute ethanol. On next day, fixed cells were centrifuged again, and cell pellet was resuspended in 1 mL PBS. Then, 100 µL of 200 µg/mL DNase-free, RNase A (Thermo Fisher Scientific, Waltham, MA, USA) was added to the cell suspension and incubated at 37 °C for 30 min. After this, 100 µL of 1 mg/mL propidium iodide (PI) was added to samples, which were incubated at room temperature for 5–10 min and then analyzed on a FACScan flow cytometer (BD Biosciences, Franklin Lakes, NJ, USA). Cell cycle analysis was carried out using Cytomation Summit Software v3.1 (Cytomation Inc., Fort Collins, CO, USA).

#### 2.2.5. Intracellular ROS Content

Cellular levels of reactive oxygen species (ROS) were measured by dyes dihydrodichlorofluorescein diacetate (DCFDA, detecting mainly H_2_O_2_) (D6883, Merck, Darmstadt, Germany) and dihydroethidium (DHE, O_2_^●−^-specific) (D7008, Merck, Darmstadt, Germany) in Jurkat T-leukemia cells. To this end, the cells were either solvent-treated or incubated with various concentrations of the new compounds and then treated with 10 μM of either DCFDA or DHE at 37 °C for 30 min. After incubation with the dyes, cells were washed with PBS and immediately analyzed with the FL1 (DCFDA) or FL2 channel (DHE) on a FACScan flow cytometer (BD Biosciences, Franklin Lakes, NJ, USA).

#### 2.2.6. Studies of the Functional Status of Mitochondria

Breakdown of ΔΨm mitochondrial membrane potential was determined by FACS analysis using 5,5′,6,6′-tetrachloro-1,1′,3,3′-tetraethylbenzimidazolylcarbocyanine iodide (JC-1) (Enzo Life Sciences, Inc., Farmingdale, NY, USA). Briefly, 10^6^ Jurkat cells were treated for 3, 6, 9, and 12 with the tested drugs. After PBS washing, cells were incubated for 10 min in freshly prepared JC-1 solution (10 mg/mL in culture medium) at 37 °C. Spare dye was removed by PBS washing and cell-associated fluorescence was measured using 605/12 nm filter (for aggregate red form) and 530/30 filter (for green monomeric form) of BD LSR Fortessa flow cytometer (BD Biosciences, Franklin Lakes, NJ, USA). Analysis of the obtained results was carried out using Cell Quest Pro software (BD Biosciences, Franklin Lakes, NJ, USA).

#### 2.2.7. FACS Analysis of Intracellular Dx Accumulation

Accumulation of free Dx, Dx-PC, and Dx-PC-NSE was measured by flow cytometry. Jurkat T-cells (5 × 10^5^ cells/well) were seeded in 24-well tissue plates (Greiner Bio-one, Frickenhausen, Germany), and after 24 h incubation were treated with various doses of studied compounds. Intracellular Dx was detected using 575/26 nm filter in BD LSR Fortessa flow cytometer (BD Biosciences, Franklin Lakes, NJ, USA), and quantified using Cell Quest Pro software (BD Biosciences, Franklin Lakes, NJ, USA).

#### 2.2.8. Microscopic Uptake Studies

HeLa cells were seeded on cover glasses in 24-well plates (Greiner Bio-one, Frickenhausen, Germany) at a concentration of 10^5^ cells per well and given 24 h to re-attach. Then, they were treated with Dx, Dx-PC or the Dx-PC-NSE complex for 6 h and stained with Hoechst 33342 (Merck, Darmstadt, Germany) for analysis of nuclear chromatin condensation. Subsequently, cells were washed in Hanks′ Balanced Salt solution (HBSS, Merck Darmstadt, Germany), and cover glasses with living cells were placed on slides. Cytomorphological studies were performed using a Carl Zeiss AxioImager A1 fluorescent microscope (Carl Zeiss, Gottingen, Germany). Image analysis was performed with Image-Pro 7.0 software (Media Cybernetics, Rockville, MD, USA).

#### 2.2.9. Immunoblot Analysis of Cellular Proteins

Western blot analysis was used to evaluate expression of proteins involved in cell death-signaling pathways in Jurkat cells after treatment with Dx, Dx-PC, or Dx-PC-NSE at various time points (3, 6, 9, 12 h). The cells were washed twice on ice with PBS before addition of lysis buffer (20 mM Tris/HCl, pH 7.5, 150 mM NaCl, 0.5% Triton X-100, 1% Trasylol, 1 mM PMSF), 20 µL per 10^6^ cells, vortexed, and centrifuged. Protein concentrations in supernatants were measured, as described [19]. Then, the appropriate volume of 5 × Laemmli buffer (10% SDS, 10% 2-mercaptoethanol, 40% glycerol, 0.01% bromphenol blue, 250 mM Tris-HCl, pH 6.8) was added, samples were heated for 5 min in boiling water and subjected to Western blot analysis. An amount of 30 µg of protein from each cell sample was loaded onto 12% polyacrylamide gels. After electrophoresis (4 h at 0.02 A), proteins were transferred onto nitrocellulose membranes (GE Healthcare Bio-Sciences, Marlborough, MA, USA) using Mini Trans-Blot Cell (Bio-Rad Laboratories, Inc., Hercules, CA) at 90 V for 1.5 h. Incubation of membranes in 5% milk solution for 1 h at 37 °C was used to block non-specific binding. Membrane was incubated with monoclonal rabbit antibodies raised against cleaved caspase-3, procaspase-6, cleaved caspase-6, cleaved caspase-7, Bid (Cell Signaling Technology, Inc., Danvers, MA, USA), AIF (sc-5586), Bcl-X_L/S_ (sc-634) (Santa Cruz Biotechnology, Inc., Dallas, TX, USA), caspase-9 (BD 556585) (BD Biosciences, San Jose, CA, USA), β-actin (Sigma-Aldrich, St. Louis, MO), and monoclonal mouse antibodies against PARP-1 (BD 556434), FADD (BD 556402), Mcl-1 (BD 559027), caspase-2 (BD 611022), caspase-8 (BD 551242) (BD Biosciences, San Jose, CA, USA), caspase-10 (M059-3) (MBL International Corporation, Woburn, MA) for 12 h at 4 °C with slow shaking. Dilution for primary antibodies was 1:1000 in 5% BSA, 0.1% PBS-Tween, except for antibodies against β-actin (1:5000), as recommended by supplier. After incubation with primary antibody, the membrane was washed three times for 5 min in PBS with 0.1% Tween 20 and then incubated in 1:5000 dilution of secondary anti-rabbit IgG horseradish peroxidase-linked antibody (GE Healthcare Bio-Sciences, Marlborough, MA, USA) for 1 h at room temperature. The membrane was washed 3 times with PBS-Tween 20 for 5 min, and proteins that bound antibodies were visualized by incubation of the membrane for 1 min in ECL buffer (1.25 mM luminol (Sigma-Aldrich, St. Louis, MO), 2.72 mM cumaric acid (Sigma-Aldrich, St. Louis, MO, USA) and 0.01% H_2_O_2_ in 0.1 M Tris-HCl (pH 8.5), and exhibition for 10–15 min to an X-ray film (Fujifilm, Tokyo, Japan). Relative amounts of protein in the electrophoretic bands were quantified by using ImageJ Analysis software (www.imagej.net, accessed on 30 November 2022). Protein normalization was conducted for β-actin levels in the same samples.

#### 2.2.10. Animals

The male Balb/c, DBA/2, and BDF1 (hybrid of Balb/c × DBA/2) mice used in this work were bred at the Institute of Cell Biology NAS of Ukraine and housed at the animal facility of the institute in a temperature-controlled environment with 12 h light/dark cycles, standard food ration for rodents (Vita Ltd., Obukhiv, Ukraine) and water *ad libitum*. Mice were checked on a daily basis for health and well-being. The procedures of all in vivo studies gained ethical approval from the BioEthics Committee of the Institute of Cell Biology, NAS of Ukraine (protocol N9/2014 dated 1.09.2014). The study on the lipid profile was carried out with the approval of BioEthics and Biosafety Committee of the Palladin Institute of Biochemistry of NAS of Ukraine (Protocol #08/09-2015).

#### 2.2.11. Murine Tumor Models and Drug Treatment Schemes

NK/Ly lymphoma cells were maintained by intraperitoneal transplantation of 0.2–0.3 mL of ascites fluid (0.5–1 million cells) on days 7–8 of lymphoma growth from a tumor-carrying animal to a recipient animal of the Balb/c strain. L1210 leukemia was passaged in hybrid BDF1 mice by injection of 1 × 10^5^ tumor cells taken from a donor animal on days 6–7 of tumor development and washed with PBS.

Animal treatment studies were done in Balb/c mice and consisted of 5 separate experiments for NK/Ly lymphoma, due to large number of tested drug concentrations. For every dose of compound (equivalent to 0.1 mg/kg, 0.2 mg/kg, 0.5 mg/kg, and 1 mg/kg Dx), animals were divided into four experimental groups with 6 animals in each. In a separate experiment, the anticancer activity of PC with or without immobilized NSE was studied in detail (n = 4 animals per experiment group). To this end, 1 mL of NK/Ly lymphoma cells in HBSS (containing about 1 × 10^6^ cells) were injected i.p. per animal. The test compounds in different doses (0.1 mg/kg, 0.2 mg/kg, 0.5 mg/kg, 1 mg/kg) were administered i.p. in a volume of 0.2 mL (dissolved in saline) per mouse every other day from the 2nd to the 20th day after tumor inoculation (control mice were treated with saline, only).

Animal studies on L1210 leukemia were done in BDF1 mice in 4 separate experiments. For every dose of compound (equivalent to 0.2 mg/kg, 0.5 mg/kg, and 1 mg/kg Dx), animals were divided into four experimental groups with 6 animals in each. In a separate experiment, anticancer activity of the PC with or without immobilized NSE was studied in detail, and in this case, animals were divided into 10 experimental groups with 4 animals in each. L1210 leukemia cells (10^5^ per animal) were inoculated i.p. to BDF1 mice, the course of treatment consisted of 8 injections of studied compounds in different doses (0.2 mg/kg, 0.5 mg/kg, 1 mg/kg), which were administered i.p. per mouse every day from 2nd to the 9th day after tumor inoculation according to standard protocol [20]. The body weight of mice was measured daily and their morphophysiological state was recorded. Overall survival analysis (Kaplan Meier curves) was performed with GraphPad Prism 8 software (GraphPad Software, Inc., San Diego, CA, USA). Comparison of survival curves was done by log-rank Mantel-Cox test.

#### 2.2.12. Blood Cell and Serum Analysis in Mice

For hematological analysis, healthy Balb/c mice (5 animals per group) were treated with 5 injections of 4 mg/kg dose of Dx i.p. in a volume of 0.2 mL (dissolved in saline) per mouse every other day (control mice were treated with saline, only). Dosages of Dx-PC and Dx-PC-NSE were re-calculated to ensure the final dose of Dx injected into mice (4 mg/kg) was equivalent to free Dx. PC and PC-NSE were injected into animals in the same dose of the polymeric carrier, as in the system loaded with Dx. Blood samples (100 μL) were collected via retro-orbital bleeding in microtubes containing 5 µL of 10% ethylenediaminetetraacetic acid disodium salt solution (Na_2_-EDTA solution) as an anticoagulant. Hematological profiles in the inspected mice, including counts for hemoglobin level, red blood cells, platelets, white blood cells, relative neutrophils (%), lymphocytes (%), and monocytes (%) were studied using a 5-Part automated hematology analyzer DF51Vet (DYMIND Biotechnology Co., Ltd., Shenzhen, China) according to manufacturer’s instructions.

For the serum biochemical analysis, blood samples were collected via retro-orbital bleeding into microtubes without anticoagulant, and serum was separated by centrifugation at 190× *g* for 10 min at 4 °C and stored in a freezer at −20 ℃ until use. Measurement of activities of aspartate aminotransferase (AST) and alanine aminotransferase (ALT), as well as concentrations of creatinine and urea, were conducted on a semi-automatic biochemistry analyzer BS-3000M (SINNOWA Medical Science & Technology Co., Ltd., Nanjing, China) using diagnostic kits from Cormay S.A. (Łomianki, Poland). All measurements were performed according to the manufacturer’s instructions.

#### 2.2.13. Analysis of Lipid Profile in Heart Tissue of Mice under Treatment with Dx and NSE

The animals were randomly divided into 5 groups, 6 animals per group. The mice of the control group received water, and the animals of the second group aqueous suspensions of the NSE at a dose of 50 mg/kg per os for 7 days. The third group of mice was treated with Dx i.p. at a dose of 2.5 mg/kg (three times with an interval of one day) in order to reach a cumulative dose of Dx of 7.5 mg/kg. The animals of the fourth group received NSE per os for 7 days at a dose of 50 mg/kg together with Dx i.p. at a dose of 2.5 mg/kg three times in an interval of one day. The fifth group of mice received NSE per os for 7 days at a dose of 50 mg/kg, after that, animals were treated with Dx i.p. at a dose of 2.5 mg/kg three times with an interval of one day. The animals of all groups were decapitated under ether anesthesia on the 12th day of the experiment. The mice’s hearts were used for analysis of the phospholipids profiles. Briefly, the content of heart phospholipids was estimated using thin-layer chromatography. The total lipids from the heart homogenates were extracted by Bligh and Dyer method [21]. The lipid extracts were stored at −20 °C in a small volume of Chl. After that, lipid extracts were separated by thin-layer chromatography on standard Sorbfil plates (Sorbpolimer, Krasnodar, Russian Federation), using solvent systems for the first dimension: Chl (65): methanol (30): ammonia (6): benzene (10), and the second dimension: Chl (5): methanol (1): acetic acid (1): water (0.5): acetone (2) [22]. Specific reagents and standards were used for identification of the individual phospholipid [23]. The individual phospholipid content was determined by Vaskovskiy and Kostetskiy [24].

#### 2.2.14. Statistical Analyses

All in vitro experiments were performed in triplicate and repeated three times. Statistical analysis of data was conducted in GraphPad Prism 8.0 software (GraphPad Software, Inc., San Diego, CA, USA) using one-way ANOVA and Tukey’s multiple comparison test or Student’s *t*-test. Statistical significance was set at *p* ≤ 0.05. Evaluation of LC_50_ values of each compound was done by built-in functions of GraphPad Prism (non-linear regression, curve fit, (Inhibitor) vs. normalized response–variable slope). For analysis of animal survival, Kaplan–Meier curves were built in GraphPad Prism, and significance to other treatments was calculated by Mantel–Cox test. Statistical analysis of biochemical indices in mice was calculated using one-way ANOVA and Tukey’s multiple comparison test.

## 3. Results

### 3.1. Assessment of Cytotoxic Activities of NSE In Vitro

First, we investigated the cytotoxic activity as well as the potential molecular mechanisms of NSE towards malignant and pseudonormal cells in vitro. NSE possessed a weak dose-dependent anticancer effect in Jurkat, L1210, MDA-MB-231 and MCF-7 cells after 24 h treatment, with an overall LC_50_ around 50 µM (Figure 3A). A potential toxicity of NAEs for breast cancer cells in dependence on a degree of their sensitivity to estrogens deserves a special attention, and the existing data on the action of these cannabinoid-like compounds are fragmentary and insufficient [25]. It was found that NSE exhibited approximately 30% higher activity towards estrogen-resistant MDA-MD-231 cells (LC_50_ = 30 μM) than in estrogen-sensitive MCF-7 cells (LC_50_ = 50 μM) (Figure 3A). These observations indicate on high promise of further in-depth studies of anticancer activity of NSE towards estrogen-resistant breast cancers.

On the contrary, human pseudonormal HEK293 cells were resistant to NSE even at high doses, which suggests a certain selectivity in the action of this cannabinoid-like compound for malignant as compared to normal cells (Figure 3A). A potential reason for this activity of NSE might be its incorporation into the plasma membrane of treated cells which affects membrane structure. This hypothesis is supported by a statistically significant increase in the size of treated Jurkat cells after exposure to high doses (30 μM) of NSE, as well as a blockade of these cells in the G_1_ phase of the cell cycle, that was demonstrated by FACS analysis (Figure 3B, Appendix A). However, no signs of apoptotic (pre-G_1_) cells were observed in these studies, despite a statistically significant decrease in Jurkat cell number under treatment with high doses of NSE (Figure 3B). In order to confirm this finding, phosphatidylserine externalization was studied by FACS using Annexin V assay at 48 h of Jurkat cell incubation with various doses of NSE. It was revealed that NSE dose-dependently decreased the number of Annexin V(+) (apoptotic) cells compared to the control, which reached the lowest value under treatment with the highest (50 µM) dose of NSE (Figure 3C). At the same time, according to the trypan blue assay, NSE (50 µM) led to a 40% decrease in the number of Jurkat cells compared to the control (Figure 3C). Taken altogether, these data clearly indicate the cytostatic mode of action of NSE, which inhibits the growth of tumor cells but does not lead to their death, and thus indirectly supports our hypothesis about incorporating this lipid into the plasma membrane of tumor cells.

Earlier, it was shown that NSE possesses both cytoprotective and antioxidant actions in vivo [26,27]. However, the mechanisms of these phenomena were not sufficiently studied. Here, we investigated the effect of this cannabinoid-like compound on the level of two main types of ROS–hydrogen peroxide and superoxide anions-in human T-leukemia Jurkat cells. To do that, flow cytometry was applied using specific fluorescent dyes–DCFDA for hydrogen peroxide, and DHE for superoxide anions. It was found that NSE did not affect the levels of both ROS types in Jurkat cells at early time points (1, 3, 6, 12 h), but led to a 20–25% decrease in the basal superoxide anion levels after 24 h incubation (Figure 3D). A similar, but less pronounced tendency was observed concerning the NSE-mediated inhibition of hydrogen peroxide production (Figure 3D).

### 3.2. Impact of NSE Pre-Treatment on the Cytotoxic Activity of Dx In Vitro

To study whether NSE is capable of modifying the cytotoxic activity of Dx, we co-exposed Jurkat cells to both drugs. We did not detect a cytoprotective effect of NSE in all concentration range towards the anticancer activity of high doses of Dx (0.5 μM, 24 h), while such effects were observed upon treatment with low Dx doses (0.25 μM, 24 h, 0.05 µM, 48 h) and high doses of NSE (20 µM) (Figure 4A). For confirmation of obtained data, Annexin V/PI double staining assay was addressed. It was revealed that co-treatment of Jurkat T-cells with NSE in high doses (30 µM and 50 µM) and low doses of Dx (0.01 and 0.05 µM, respectively), led to a profound decrease in the number of apoptotic cells at 48 h incubation. However, no anti-apoptotic effect of NSE was observed under cell treatment with high doses of Dx (0.25 µM and 0.5 µM) (Figure 4B). Next, we aimed to study if the detected cytoprotective actions of NSE) could be caused by the intrinsic antioxidant activity of this lipid, or whether it is more likely due to the incorporation of NSE into the plasma membranes of treated cells, which may physically affect the penetration of the Dx into the target cells.

To explore this effect in more detail, we measured the level of H_2_O_2_ and O_2_^−^ in Jurkat cells treated with LC_50_ dose of Dx (0.5 µM) in combination with NSE in different doses (10 μM, 20 μM and 30 µM). However, no significant inhibitory effect of NSE on hydrogen peroxide or superoxide anion levels, considered as the main cause of Dx-induced cardiotoxicity, were found (Figure 4C). Thus, we assume that the cytoprotective activity of NSE towards low doses of Dx at late time points is caused predominantly by its membrane-binding properties rather than antioxidant activities.

### 3.3. Modulation of the Cytotoxic Effect of Dx towards Tumor Cells In Vitro by NSE Immobilized on PC

In part 3.2 of the Results section, we showed that NSE was not a very effective modulator of Dx action in vitro. Thus, we immobilized NSE on a new polymeric carrier based on poly(5-(tert-butylperoxy)-5-methyl-1-hexen-3-yn-co-glycidyl methacrylate)-graft-PEG and used this formulation as a platform for targeted Dx delivery.

We explored the cytotoxic action of the Dx-polymer composite functionalized with NSE on a panel of human tumor cell lines in vitro. As shown in Table 2, co-immobilization of Dx and NSE on PC led to a marked increase in cytotoxic activity towards various tumor cell lines. In more detail, LC_50_ values of Dx-PC-NSE conjugates were found to be 2-25-fold lower than the one of free Dx and 1.5-10-fold lower than those of the Dx-PC complex (Table 2). However, the most prominent results were obtained towards tumor cell lines with MDR phenotype, characterized by overexpression of ABC-transporter proteins (Figure 5). In particular, the sensitivity of SW1573/2R160 cells (ABCB1+) to Dx-PC-NSE was 10-fold higher (LC_50_ = 0.85 µM) compared to Dx (LC_50_ = 11.02 µM), while immobilization of Dx on PC without NSE had no impact on its MDR-circumventing properties (LC_50_ = 9.32 µM). In order to verify these data, HL-60/adr (ABCC1+) cells were used delivering similar results with the Dx-PC-NSE complex (Figure 5, Table 2).

Circumvention of cancer drug resistance may be achieved either by inhibition of ABC transporter activity using specific small molecules (e.g., verapamil) or by the enhanced entry of an anticancer drug into the cytoplasm or nuclei in a way not vulnerable to ABC transporter-mediated efflux from the membrane bilayer.

In order to verify whether NSE is really capable of facilitating the entry of the Dx-polymeric carrier to treated cells, a cytomorphological study on HeLa cells was carried out. It was shown that after 6 h, Dx-associated fluorescence (2 μM) was detected in the cytosol of treated cells, but not in the nuclei (Figure 6). We observed a beginning of the condensation of chromatin in these cells without other pronounced cytomorphological signs of apoptosis, such as a formation of protrusions on the surface of the cell membrane or apoptotic bodies. An increase in the Dx concentration to 4 μM further enhanced its penetration into nuclei of target cells and the manifestation of the apoptotic cell death by the formation of apoptotic bodies.

The immobilization of the Dx to PC enhanced drug penetration into the nucleus of the target cells, but the effect was well seen only at a dose of 4 μM (Figure 6). Here, distinct apoptotic changes in the cell morphology were detected, such as membrane blebbing, formation of apoptotic bodies, etc. When NSE was included in the Dx delivery system (PC with 3% *w*/*v* both of Dx and NSE, molar ratio Dx: NSE 1: 1.82), the drug fluorescence was detected in the nuclei of treated cells even at the lower dose of 2 μM. Thus, the presence of NSE in the composition of the Dx-loaded PC significantly (2-fold) enhanced the penetration of this drug to the nucleus more efficiently than the carrier itself. This is well reflected by the decreased LC_50_ values of Dx-PC-NSE in comparison to Dx and Dx-PC (Table 2).

We hypothesized that the higher lipophilicity of PC-NSE compared to non-functionalized PC is the main reason for the rapid entry of Dx, immobilized on PC-NSE, inside tumor cells (including those with MDR phenotype) and thus increased apoptosis induction. In order to confirm this hypothesis, another *N*-acylethanolamine lacking two methylene groups in its lipid chain (NPE, *N*-palmitoyilethanolamine), was prepared. Subsequent biological experiments showed that the activity of this Dx-PC-NPE complex (16 methylene groups in lipid chain) towards MCF-7 cells was 20–30% lower than that of Dx-PC-NSE (18 methylene groups in lipid chain) (Appendix A). This effect was also observed in L1210 cells, but only at higher concentrations of Dx (2 µM). Thus, the length of the fatty acid chain in the N-acylethanolamine molecule, immobilized to PC, plays an important role in the enhancement of the entry of anticancer drugs into tumor cells.

In order to reveal if this phenomenon is somehow connected to enhanced entry of NSE-functionalized polymeric carrier with Dx into the cells or to enhanced Dx release by PC-NSE, we have performed quantitative analysis of Dx release from PC and PC-NSE in cell-free systems and FACS studies of Dx entry into Jurkat T-cells in time-dependent mode. As one can see in Figure 7A, free Dx diffuses through the dialysis membrane with a rather high velocity, and the equilibrium was achieved in approximately 240 min. Dx release from its complex with PC was much less intensive, compared with a release of a free drug. Up to 75 min, the dynamics of the release of the Dx from its complex with PC-NSE did not differ significantly from such dynamics of its release from a non-functionalized PC. However, after 75 min, a release of the Dx from its complex with PC-NSE was considerably accelerated, and it practically reached a release intensity of free Dx. However, FACS studies have revealed even more rapid dynamics for Dx-PC-NSE entry into Jurkat cells (Figure 7B), which was 15–20% faster to Dx and Dx-PC at 30 min and 60 min time points, but significantly accelerated at 90 min and 120 min at higher concentrations, reaching equilibrium at 6 h (Dx 4 µM) and 12 h (Dx 2 µM) (see Appendix A). These results clearly indicate that the increase in the cytotoxic activity of Dx-PC-NSE in vitro is caused by two different mechanisms: enhanced entry into target cells due to increased lipophilicity of the drug delivery system (stage 1, up to 90 min) and then enhanced release of Dx in the cytosol of tumor cells by PC-NSE (stage 2, from 90 min till 360–720 min depending on Dx concentration). In all cases the efficiency of Dx entry and release by the Dx-PC complex (without NSE) was lower compared to Dx-PC-NSE, thus confirming the crucial role of NSE in this phenomenon.

### 3.4. Impact of Co-Immobilization of Dx and NSE on PC on ROS and Induction of Cell Death in Tumor Cells In Vitro

In order to investigate the impact of enhanced Dx delivery into tumor cells by NSE-functionalized PC on the Dx mode of action, treatment-induced changes in cell cycle distribution, ROS levels, functional status of mitochondria, and the activation of apoptosis-promoting signaling pathways were studied in more detail. Enhanced entry of Dx into the tumor cells, driven by NSE-containing polymers, also induced significantly altered cell cycle distribution (Figure 8A). While Dx at 0.1 µM led to a moderate increase in the G_2_ peak, Dx-PC-NSE in the same concentration caused a complete G_2_-block in Jurkat cells. The same effect was achieved when Jurkat cells were treated with a five-fold higher dose of Dx (0.5 µM). However, when the Dx-PC-NSE complex was used at equimolar concentrations, it led to further changes in cell cycle distribution, primarily an increase in the S-phase fraction (Figure 7C).

Interestingly, this strong impact of NSE on Dx-induced G_2_-arrest was also accompanied by a decrease in superoxide radical levels at early time points (1 h), which was observed at both low (2 µM) and high Dx doses (4 µM) in Jurkat cells (Figure 8B). As superoxide radicals are mainly produced by mitochondria, a partial decrease in their production by Dx-PC-NSE conjugates may be caused by the embedding of NSE into the outer mitochondrial membrane induced by the carrier and, thus, stabilization of mitochondrial functions for a limited period of time. These tendencies were also observed at late time points (3, 6, 12 h), but without statistical significance (Figure 8B). No changes in the levels of hydrogen peroxide were observed in cells treated with Dx or Dx-PC-NSE at all studied time points (Figure 8C). As H_2_O_2_ in mammalian cells is mostly produced by cytosolic NADPH dehydrogenases [28], loss of its fluctuations by Dx-PC-NSE clearly indicates partial inhibition of superoxide production in mitochondria due to the antioxidant features of NSE. For confirmation of this hypothesis, JC-1 staining of Jurkat cells under the treatment with Dx, Dx-PC, and Dx-PC-NSE was carried out (Appendix A). Surprisingly, no changes in mitochondrial membrane potential were revealed by us either at 3 h, or at 6 h time point, while at 9 h a weak depolarization of mitochondria was observed under Dx treatment (2 µM), which was partially decreased under treatment with Dx-PC-NSE, but not Dx-PC. However, the use of a higher dose of Dx (4 µM) led to an actual decrease in depolarized mitochondria compared to a lower dose (2 µM) at 9 h and 12 h, despite massive cell death, observed by trypan blue assay (Appendix A). We assume that in high concentrations Dx leads to mainly DNA intercalation instead of redox-cycling, which explains the weak involvement of mitochondria in our case. In order to confirm these data, a Western blot analysis of time-dependent activation of signaling pathways of apoptosis in Jurkat cells under the treatment with Dx, Dx-PC, and Dx-PC-NSE was carried out (Figure 9). Dx treatment (2 µM) led to weak cleavage of initiator caspases-8, -9, -10 at all time points, thus confirming JC-1 data about insignificant involvement of mitochondria in Dx-induced cell death at high doses of this drug. Application of the Dx-PC-NSE led to significantly later (9 h) activation (cleavage) of initiator caspase-9, involved in mitochondria-mediated apoptosis, in contrast to Dx and Dx-PC, where this effect was observed as soon at 3 h (Figure 8, Appendix A). On the contrary, activation of the effector caspase-3 was stronger for Dx-PC-NSE compared to Dx and Dx-PC. This may indicate a specific switch to mitochondria-independent apoptosis upon treatment with Dx-PC-NSE due to partial inhibition of mitochondria-driven ROS production. At later time points (12 h), Dx-PC-NSE led to full (apoptosis-induced) cleavage of PARP-1 and DFF45, procaspases-2,-8,-10, on contrary to Dx and Dx-PC used in the same concentration (2 µM). These results represent a further confirmation of an altered and more potent proapoptotic potential of Dx-PC-NSE towards tumor cells in vitro.

### 3.5. Activity of PC with Immobilized Dx and NSE in Tumor-Bearing Mouse Models

In our study, two experimental models of myeloid tumors in mice were used-NK/Ly lymphoma and L1210 leukemia which differ significantly in the speed of cancer development after tumor inoculation as well as resistance to anticancer drugs [29,30,31]. Dx in low doses (0.1 mg/kg, 10 injections from 2 to 20 days after tumor inoculation) demonstrated a weak therapeutic effect with an increase in the average life span in NK/Ly lymphoma-bearing animals from 21 to 36 days, while Dx-PC, on contrary, accelerated the death of animals (decreased from 21 to 16 days). This effect might be explained by the fact that immobilization of the Dx on PC simultaneously enhanced its antitumor effect as well as negative side effects in tumor-bearing animals leading to their rapid death. Functionalization of the Dx-PC with NSE had a positive effect since an increase in the average lifespan from 21 to 24 days was observed and some animals lived several days longer than animals of the Dx-treated group (Figure 10A). Use of a higher dose (0.2 mg/kg) of the Dx demonstrated significantly better therapeutic indicators: an increase in the life expectancy of the Dx group from 21 to 40.5 days, the Dx-PC group from 21.5 to 34 days, and a further increase up to 49 days in Dx-PC-NSE group. The maximum lifespan of animals treated with the Dx-PC-NSE was 50 days, while in mice treated with free Dx the overall survival did not exceed 45 days.

Thus, even low doses of Dx co-immobilized with NSE on PC demonstrated a pronounced enhancement of the therapeutic effect in comparison to Dx in its free form. A further increase in the Dx single injection dose to 0.5 mg/kg was still insufficient for the complete cure of animals (median survival 51.5 days and maximum survival 67 days). However, in the Dx-PC group, only 16% of animals were alive 90 days after treatment, while in the Dx-PC-NSE group, all animals exhibited no signs of the tumor until the end of the experiment, thus indicating a cure from this tumor. The same effect with Dx used alone was observed only when a cumulative dose of 10 mg/kg was applied (Figure 10A). Thus, the application of Dx-PC-NSE induced complete recovery of mice from NK/Ly lymphoma in half of the cumulative dose of free Dx–5 mg/kg versus 10 mg/kg. These results are in agreement with our data obtained in the in vitro studies, in which Dx-PC-NSE demonstrated at least two-fold higher activity than free Dx.

In order to confirm the results obtained with NK/Ly lymphoma, another murine tumor model—L1210 leukemia—was used. In these experiments, a dose of 0.2 mg/kg Dx was insufficient to achieve a therapeutic effect, although even in this case, treatment of the animals with Dx-PC-NSE led to an improvement of their lifespan from 18 to 22 days compared to the effect of free Dx. When a 0.5 mg/kg dose of Dx was used, the lifespan of animals increased proportionally from 10.5 to 23 days, with maximum overall survival of 24 days. Dx-PC-NSE increased the life span to 33 days and 33% of animals showed signs of complete clinical response (life expectancy > 90 days) (Figure 10B). In the groups treated with 1 mg/kg Dx (cumulative dose of 8 mg/kg), there was no cure for tumor-bearing animals, although their life duration was increased from 10.5 to 31 days (Figure 10B). Dx, immobilized on PC increased the life span by only 3 days, while treatment of animals with Dx-PC-NSE at a similar dose resulted in complete remission of all treated animals (survival >90 days) (Figure 10B). In order to ensure that the observed phenomenon is not caused by innate anticancer activity PC or PC-NSE, their impact on the same doses, used for delivery of Dx (0.1 mg/kg, 0.2 mg/kg, 0.5 mg/kg and 1 mg/kg) to mice were addressed in separate studies on NK/Ly lymphoma and L1210 leukemia. We did not observe any statistically significant impact of both low and high doses of PC and PC-NSE on the survival of tumor-bearing animals (Appendix A), which indicates the absence of therapeutic activity of PC or its conjugate with NSE.

Summarizing, co-immobilization of Dx and NSE on PC leads to an approximately two-fold increase in the therapeutic activity of Dx in two different experimental murine tumor models, leading even to complete remissions in several animals. The main reason for the observed effect is the co-immobilization of the NSE and Dx on PC since the therapeutic activity of Dx-PC only slightly differs from such activity of the Dx used alone.

### 3.6. Bio-Protective Effects of NSE Conjugated to PC on General Toxicity of Dx In Vivo

In order to reveal if binding of Dx to NSE-polymer complexes, besides enhancement of therapeutic activity, may also lead to a decrease in its main side effects (e.g., myelosuppression, hepato-, cardio- and nephrotoxicity), blood analysis of healthy Balb/c mice treated with sub-lethal dose of Dx (20 mg/kg) was carried out in a time-dependent manner. Dx led to a two-fold decrease in the white blood cell count already 15 days after treatment, that was accompanied by neutrophilia and lymphocytopenia (Figure 11). Immobilization of Dx to PC prohibited Dx-induced lowering of WBC levels in the treated mice. However, we observed simultaneous increase in level of monocytes, which is considered a marker of prothrombotic activity of Dx [32], thus, indicating an increased risk of thrombosis. On contrary, Dx-PC-NSE had no impact on level of monocytes, but effectively prohibited Dx-induced decrease in WBC number altogether with neutrophil and lymphocyte levels, thus, demonstrating much complex and better protection of hematopoietic system of mice towards Dx.

No statistically significant changes in the levels of creatinine and urea were observed in this experiment, indicating the absence of considerable kidney damage by the compounds at the concentrations used. However, AST activity was significantly elevated in the blood of Dx-treated mice, suggesting severe liver and heart damage. In contrast, Dx-PC-NSE did not induce these negative side effects of Dx (Figure 12).

In another experiment, the phospholipid profile in the heart tissue of mice under treatment with Dx and/or NSE was investigated. A significant decrease in the content of cardiolipin (DPG) and sphingomyelin (SM) was detected, while the levels of the phosphatidylinositol (PI), phosphatidylserine (PS), and lysophosphatidylcholine (LPC) were considerably increased under by Dx treatment. No changes were detected in the content of the phosphatidylcholine (PCh) and phosphatidylethanolamine (PE) under treatment with Dx (Table 3). It should be stressed that in all cases, when the content of specific phospholipids was disturbed by Dx, NSE demonstrated a distinct normalizing effect by preventing a decrease in DPG and the SM and an increase in PS, PI, and the LPC. The administration of NSE before Dx prevented the DPG decrease as well as the increase in PI and PS. However, the administration of the NSE did not have a significant effect on the SM and LPC content. Administration of the NSE without Dx did not affect the profile of phospholipids in the heart tissue of mice.

Thus, we demonstrated that the application of Dx-PC-NSE at least partly prevents the negative side effects caused by the Dx in a short time interval, supporting our initial hypothesis about the dual role of NSE in mediating the anticancer potential of Dx. On the one hand, the lipophilicity of the NSE may speed up the entry of Dx inside tumor cells and nuclei even in MDR cells, leading to enhanced apoptosis induction. On the other hand, the tissue-protective activity of NSE would lead to better protection of the liver and hematopoietic system of mice at sub-lethal concentrations of Dx.

## 4. Discussion

Up to now, Dx remains one of the most widely used anticancer drugs for the treatment of cancer of different types and tissue origins [33]. However, its use in clinical practice is limited due to the cumulative lifetime cardiotoxicity, which is mainly caused by the excessive production of superoxide anions in cardiomyocytes [34], and rapid development of ABC transporter-mediated resistance in tumor cells [35,36]. In order to overcome these limitations of Dx, two main approaches are used—chemical modification of its molecule and nanocarriers for its delivery through the organism. In the first approach, its stereoisomer, epirubicin, was synthesized, which is characterized by 10-fold longer half-life and increased volume of distribution. In addition, elimination of 4-methoxy group in ring D was shown to lead to a major increase in a lipophilicity of the idarubicin, thus, enhancing its uptake into the targeted cells [34]. However, these chemical modifications of the anthracyclines led mainly to an increase in their cytotoxic activity, with a moderate effect on their adverse effects in the body of cancer patients [37,38]. On contrary, liposome-based nano-formulations of Dx, such as Doxil^®^ (~100 nm, PEG-functionalized) or Myocet^®^ (liposome-based non-PEGylated Dx) have shown significantly decreased side effects, compared to original Dx, but there was no distinct increase in its therapeutic efficiency [39]. That is why the development of an advanced delivery system for Dx capable of a simultaneous increase in its therapeutic activity and lowering its adverse effects, remains a crucial task in cancer chemotherapy.

In previous studies, we revealed that the use of synthetic comb-like PEGylated polymers reduced the effective dose of Dx but had little effect on its adverse effects in vivo [7]. Here, we proposed to combine both aforementioned strategies—simultaneous enhancement of lipophilicity and an increase in bio-distribution of Dx through its co-immobilization on a PC conjugated with a synthetic cannabimimetic NSE possessing innate anti-inflammatory and antioxidant activity.

Earlier, it was demonstrated, that NSE administration per os led to a partial normalization of the activities of superoxide dismutase and glutathione peroxidase in heart tissue of rats, which were severely decreased under Dx treatment [40]. Similar tissue-protective effects of the NSE were observed in mice bearing Lewis lung carcinoma, which led to partial normalization of creatinine and ALT levels in the blood, elevated under the Dx treatment [41]. However, up to now, there was no information about the potential impact of NSE on the anticancer activity of Dx in vivo. The main reason for that may be the poor bioavailability of NSE administered per os. In order to circumvent this problem and increase NSE availability, we aimed to co-immobilize this agent on PC.

To select proper concentrations of NSE for PC immobilization, we performed in vitro studies of its cytotoxic activity towards several tumor cell lines as compared to pseudonormal cells. NSE possesses weak cytotoxic activity towards tumor cells with an average LC_50_ in the 30–50 µM range depending on the cell line, while pseudonormal human kidney embryonal HEK293 cells were found to be more resistant to NSE’s action. This indicates the selectivity of NSE towards malignant cells in vitro. Inhibition of tumor cell growth by NSE was accompanied by an increase in cell size as well as cell cycle arrest in the G_1_ phase. Despite dose- and time-dependent decrease in cell number under NSE treatment, it led to almost complete inhibition of basal level of apoptosis in the population of Jurkat cells at 48 h. Thus, we suggest that the main mode of cytostatic action of the NSE in vitro might be based on the enhanced incorporation of this lipid-like compound into the plasma membrane of target cells that may lead to inhibition of their proliferation. NSE also demonstrated a weak antioxidant activity at 24 h treatment of Jurkat T-leukemia cells, but this effect was not statistically significant.

Co-treatment of human Jurkat T-leukemia cells with various doses of Dx and NSE has revealed the weak antagonistic effect of these compounds only at low doses (0.01 µM, 0.05 µM) for 48 h, revealed by trypan blue exclusion assay and Annexin V/PI double staining. We did not observe any impact of the NSE on the levels of hydrogen peroxide and superoxide radicals in Jurkat T-cells when applying an LC_50_ dose (0.5 µM) of Dx. Thus, we assume that the antioxidant activity of the NSE per se is not sufficient to modulate its action towards Dx-induced redox cycling. That is why further efforts were made to increase the bio-availability of the NSE via its immobilization to PC together with the Dx. Several formulations with a variable ratio of Dx and NSE were synthesized, and the most stable composite with 1 part of Dx and 1 part of NSE w/w (equivalent to 1:1.82 molar ratio) was selected for further studies.

We detected that co-immobilization of Dx and NSE significantly increased the potential lipophilicity of Dx-PC-NSE, resulting in its enhanced uptake in vitro. Moreover, drug release studies had revealed an increased release rate of Dx from PC-NSE, starting at 75 min timepoint, which resulted in a further boost of Dx entry inside Jurkat cells at 90 min, as revealed by FACS analysis of Dx uptake. This, in turn, led to an elevated proapoptotic and cytotoxic activity of Dx towards a series of tumor cell lines, and also is suggested the main reason for partial circumvention of ABCC1 and ABCB1-driven resistance of cancer cells in vitro by Dx-PC-NSE, but not Dx-PC. On the other side, immobilization of the NSE to PC also led to enhanced antioxidant properties. The increase in the anticancer activity of Dx-PC-NSE complexes was accompanied by a significant decrease (*p* < 0.05) of superoxide production in Jurkat T-leukemia cells at early time points after treatment with various doses of Dx-PC-NSE, but not with Dx and Dx-PC. This may suggest partial inhibition of Dx-induced mitochondrial depolarization by NSE at early time points that might be a reason for its antioxidant activity in vivo, observed by us earlier [40]. However, JC-1 studies had revealed very weak depolarization of mitochondria in Jurkat cells, treated with high doses of Dx, suggesting little involvement of Dx-induced redox cycling at these drug concentrations. Nevertheless, PC-NSE was found to decrease the number of depolarized mitochondria at the 9 h time point, thus confirming our suggestion about the mitochondria-stabilizing function of NSE. Western blot analysis of key proteins involved in early and late stages of apoptosis demonstrated that activation of the initiator caspase-9 was delayed by treatment of Dx-PC-NSE, while the level of cleaved (active) caspase-3, on contrary, was increased, compared to Dx and Dx-PC. However, at late time points (12 h) activation of all studied caspases and cleavage of their substrates was significantly stronger for Dx-PC-NSE as compared to Dx and Dx-PC. These results suggest partial loss of mitochondrial apoptosis induction in Jurkat T-leukemia cells by Dx-NSE-PC but confirm much stronger induction of apoptosis by this composite at later time points.

Therapeutic effects of Dx-PC-NSE were compared to free Dx and unloaded PC by application to mice carrying NK/Ly lymphoma or L1210 leukemia grafts. We observed a distinct dose-dependent enhancement of the therapeutic activity of Dx towards NK/Ly lymphoma by co-immobilized NSE. These data also suggest a crucial role of the NSE in the modulation of side effects of Dx in vivo.

As predicted, L1210 leukemia was found to be more resistant to Dx treatment. However, the application of Dx-PC-NSE composite with a Dx concentration equal to 1 mg/kg (8 mg/kg cumulative dose) led to 100% recovery of all mice in the experimental group, thus, confirming our data, obtained from the NK/Ly lymphoma model.

In order to reveal potential mechanisms of an enhancement of Dx therapeutic action by the NSE immobilized on the polymeric carrier, we conducted in-depth functional studies of the hematopoietic system, as well as liver, heart, and kidneys of healthy mice treated with sub-lethal doses of free Dx (20 mg/kg) and immobilized to PC with or without the conjugated NSE. Dx treatment caused severe leucopenia, thrombocytosis, lymphopenia, and neutrophilia, while the application of Dx-PC-NSE was capable of partial reduction of these adverse effects. Surprisingly, Dx-PC led to an even stronger increase in monocyte levels than free Dx, which may explain a decreased survival of tumor-bearing animals treated with Dx-PC due to the development of thrombosis [32]. Recent studies indicate that neutrophils may play a crucial role in Dx-induced cardiotoxicity by acute infiltration into the heart [42]. Thus, the normalization of neutrophil levels of Dx-PC-NSE, observed by us, may explain the absence of delayed toxicity of Dx even in long-range studies.

We also observed a two-fold elevation of AST activity (a marker of hepatotoxicity) in blood serum by treatment with a 20 mg/kg dose of Dx, while the application of Dx-PC-NSE did not induce liver toxicity. The same effect, but to a lower extent was observed for ALT, which is a known biomarker of the functional status of the liver. The absence of statistically significant changes in creatinine and urea levels in the blood of mice under these treatments is in agreement with previous data on the tissue-protective activity of the NSE [40,41,43,44].

It is known that the cardiotoxic effect of Dx is related to its high affinity to DPG, a unique phospholipid located in the inner membrane of mitochondria [45]. This lipid plays an important role in energy metabolism via providing stability for complexes of the enzymes involved in energy generation. In addition to its role in mitochondrial bioenergetics, DPG is capable of electrostatic anchoring of cytochrome c in the inner membrane of mitochondria, thus, playing an important regulatory role in a release of cytochrome c to cytosol as a key process in the initiation of apoptosis. Since the cardiomyocytes are extremely rich in mitochondria compared to other cell types, the cardiotoxicity of the Dx belongs to the earliest and most intensive manifestations of its adverse effects. We found that NSE definitely normalized changes in the phospholipids’ profile caused by the Dx in the heart tissue of mice. Taking into account the special role of the DPG in the functioning of mitochondria, such a preventive role of NSE towards the cardiotoxicity of Dx might be principal to the cardioprotective effect of the NSE.

Summarizing, co-immobilization of the NSE and Dx on a micellar polymeric carrier leads to a distinct increase in Dx uptake into tumor cells with an enhancement of the anticancer activity both in vitro and in vivo. In parallel, an increase in the antioxidant activity of the NSE takes place, when it is applied in a composite with this drug delivery platform. Such composite is capable of effective protection of heart and liver tissues of mice with transplanted NK/Ly lymphoma and L1210 leukemia from adverse effects of Dx, finally leading to the final cure of treated animals. Altogether, the obtained data suggest a high potential of Dx-NSE nanoconjugates in clinical oncological practice.

## 5. Conclusions

A unique bi-functionality of the synthetic cannabimimetic NSE was revealed. It demonstrated a proapoptotic activity on tumor cells in vitro and suppressed tumor growth in vivo. At the same time, in complex with the polymeric carrier, NSE showed a bioprotective effect against the general toxicity of Dx and particularly its cardiotoxicity. Potential mechanisms responsible for such a bioprotective role of NSE were elucidated.

## Figures and Tables

**Figure 1 pharmaceutics-15-00835-f001:**
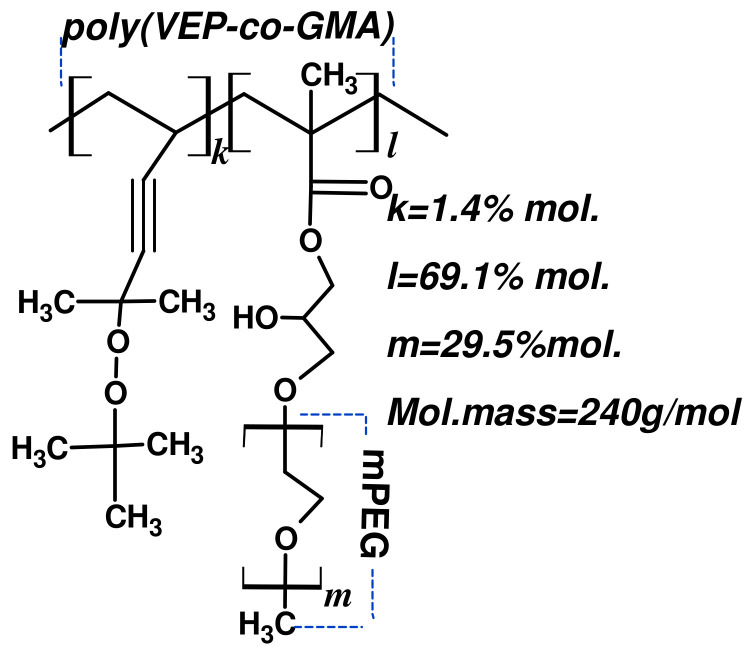
Assumed structure of the poly(VEP-co-GMA)-graft-mPEG.

**Figure 2 pharmaceutics-15-00835-f002:**
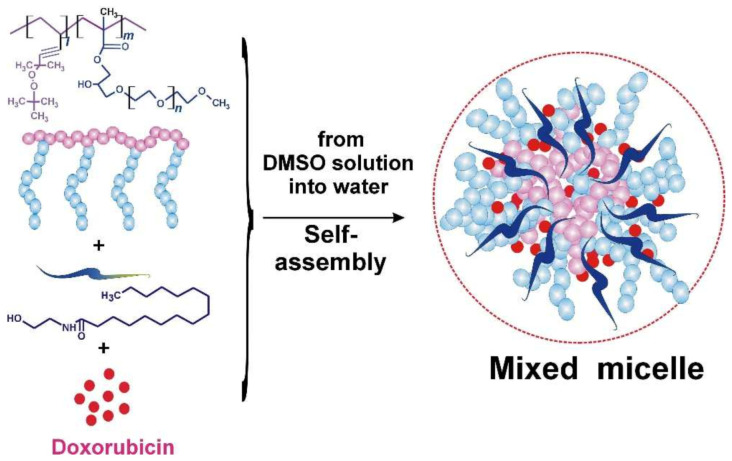
General scheme of the formation of a complex with Dx in mixed micelles with PC + NSE.

**Figure 3 pharmaceutics-15-00835-f003:**
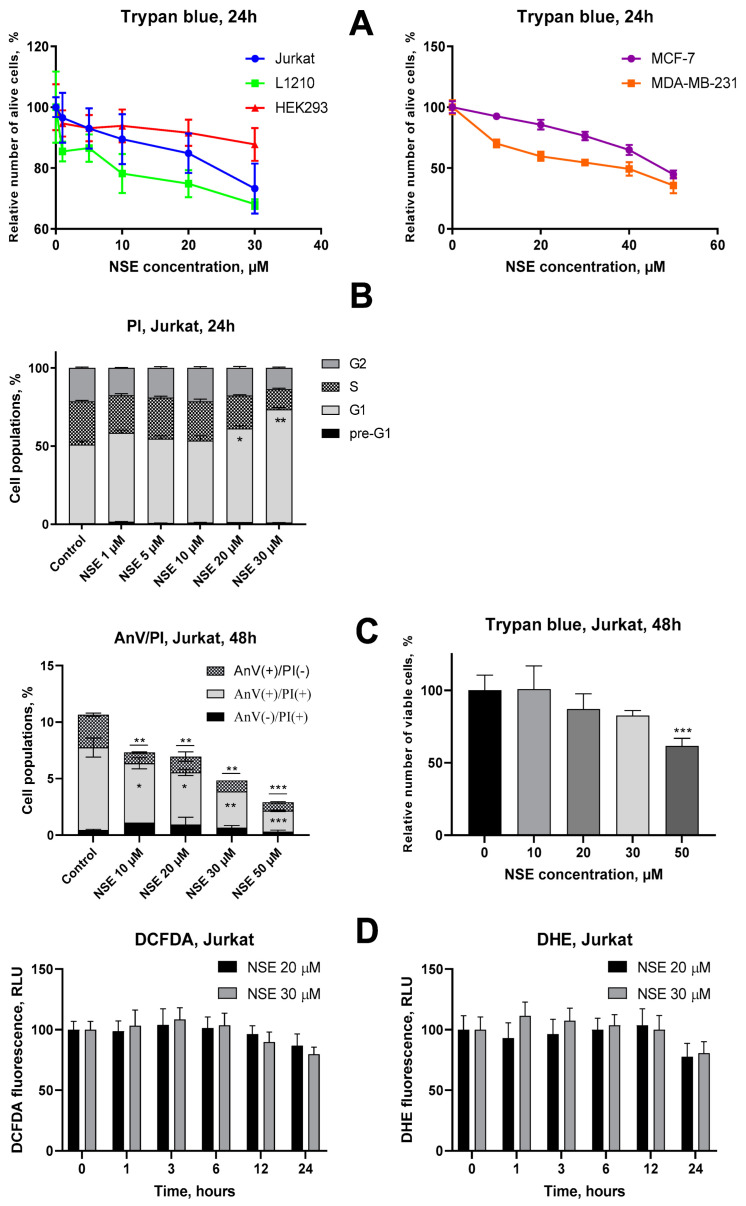
Evaluation of cytostatic and cytotoxic activities of NSE in vitro. (**A**) Viability of the indicated cell lines after treatment with various concentrations of NSE for 24 h, analyzed by trypan blue exclusion. (**B**) Changes in cell cycle progression in Jurkat cells under treatment with various concentrations of NSE, flow cytometry, and PI staining (24 h incubation). (**C**) Impact of NSE on apoptosis induction and viability of Jurkat T-cells at 48 h incubation, flow cytometry, Annexin V/PI staining, and trypan blue exclusion assay. (**D**) Time-dependent changes in H_2_O_2_ and O_2_^−^ production in Jurkat cells under treatment with various concentrations of NSE. Flow cytometry, DCFDA, and DHE staining, respectively. The effect of NSE on cell growth and ROS production was plotted relative to the untreated control. Data given represent the mean ± SD of three independent experiments performed in triplicates. Significance to control and other groups was calculated with one-way ANOVA and Tukey’s multiple comparison test (* *p* ≤ 0.05, ** *p* ≤ 0.01, *** *p* < 0.001).

**Figure 4 pharmaceutics-15-00835-f004:**
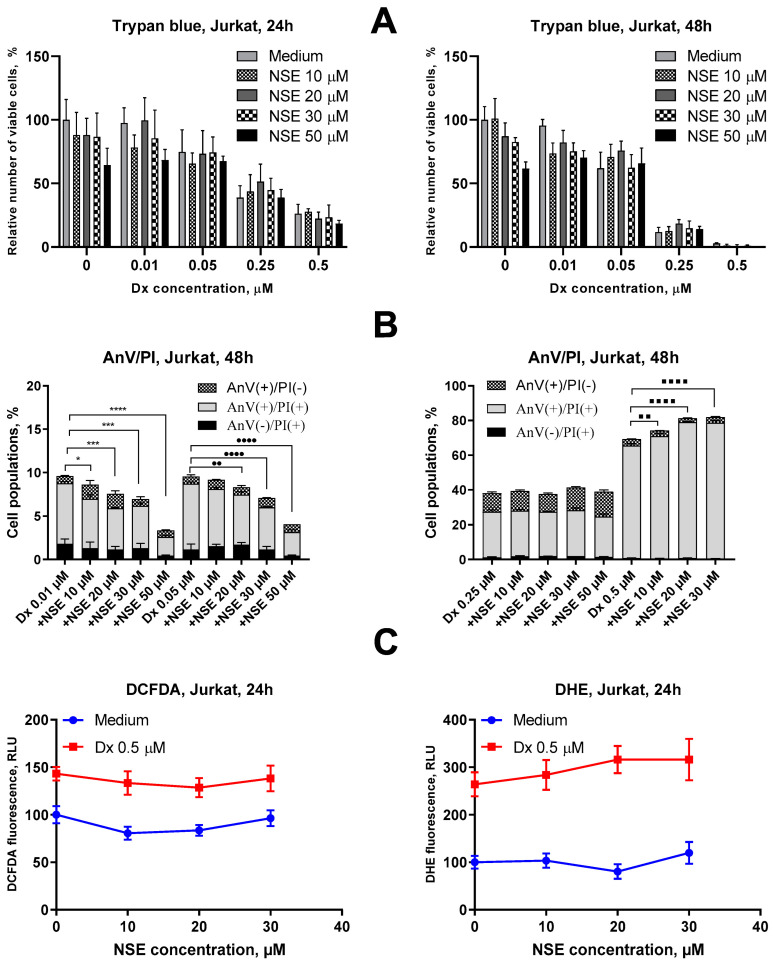
Impact of the combination of NSE and Dx in vitro. (**A**) Viability of Jurkat cells after treatment with various concentrations of Dx and/or NSE for 24 h and 48 h, analyzed by trypan blue exclusion. (**B**) Phosphatidylserine externalization in Jurkat cells after treatment with various concentrations of Dx and/or NSE for 48 h. Flow cytometry, APC-annexin V, and PI double staining. (**C**) Impact of co-treatment with NSE and Dx on H_2_O_2_ and O_2_^−^ levels in Jurkat cells. Flow cytometry, DCFDA, and DHE staining, respectively, 24 h incubation. The effect of NSE and/or Dx on cell growth and ROS production was plotted relative to the untreated control. Data given represent the mean ± SD of three independent experiments done in triplicates. Significance to control and other groups was calculated with one-way ANOVA and Tukey’s multiple comparison test (* *p* ≤ 0.05, *** *p* < 0.001, **** *p* < 0.0001—differences in number of AnV(+)/PI(+) cells relative to Dx 0.01 µM, •• *p* ≤ 0.01, •••• *p* < 0.0001 relative to Dx 0.05 µM, ▪▪ *p* ≤ 0.01, ▪▪▪▪ *p* < 0.0001 relative to Dx 0.5 µM).

**Figure 5 pharmaceutics-15-00835-f005:**
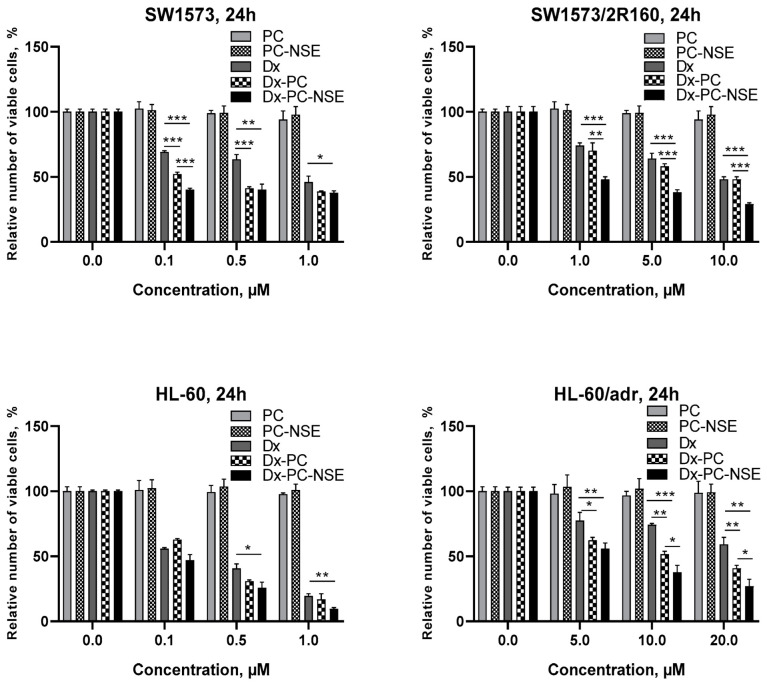
Comparison of the cytotoxic activity of polymeric micelles with immobilized Dx and/or NSE in vitro towards human SW1573 non-small lung cancer cells and its drug-resistant subline SW1573/2R160 (ABCB1+), and human leukemia HL-60 together with its Dx-resistant subline HL-60/adr (ABCC1+). The effect of studied compounds on cell growth was plotted relative to the untreated control. Data given represent the mean ± SD of three independent experiments performed in triplicates. Significance to other treatments was calculated by Student’s unpaired *t*-test (* *p* < 0.05; ** *p* < 0.01; *** *p* < 0.001).

**Figure 6 pharmaceutics-15-00835-f006:**
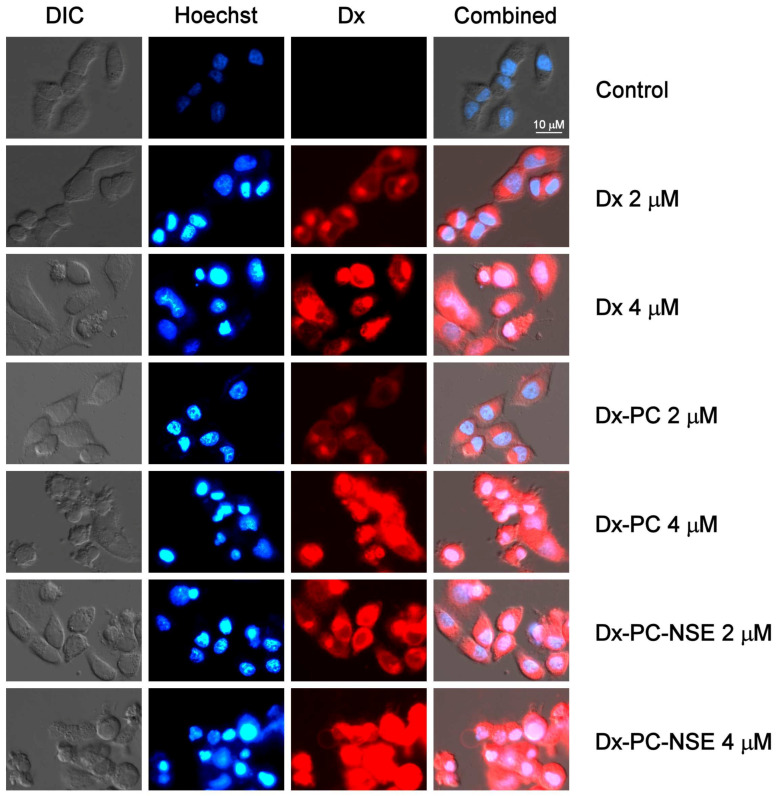
Changes in nucleus ultrastructure and Dx accumulation in HeLa cells under treatment with the indicated concentrations of Dx, Dx-PC, Dx-PC-NSE. Hoechst 33342 staining, fluorescent microscopy, 6 h incubation, ×40.

**Figure 7 pharmaceutics-15-00835-f007:**
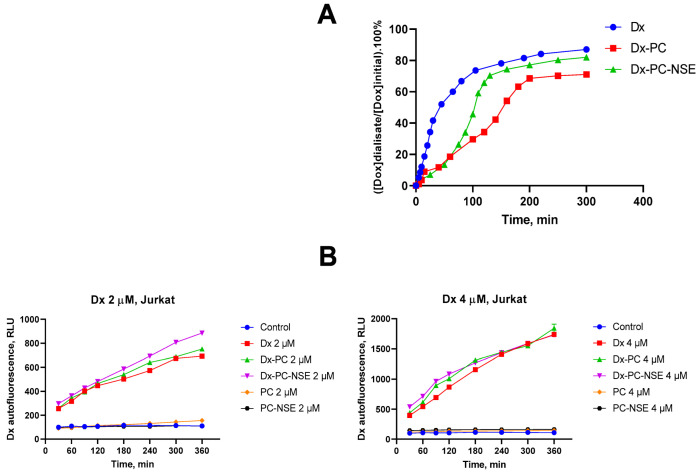
Quantitative analysis of Dx release from its complex with polymeric carrier and its entry into Jurkat T-cells. (**A**) Time-dependent dynamics of a release of free Dx (1) and of Dx immobilized in the micellar complexes Dx-PC (2) and Dx-PC-NSE (3) through the dialysis membrane in aqueous solution (initial concentrations of components loaded into the dialysis tubings were: [NSE] = 0.3 mg/mL; [PC] = 10 mg/mL; [Dx] = 0.3 mg/mL). (**B**) FACS analysis of time- and concentration-dependent dynamics of Dx, Dx-PC and Dx-PC-NSE entry into Jurkat T-cells in vitro.

**Figure 8 pharmaceutics-15-00835-f008:**
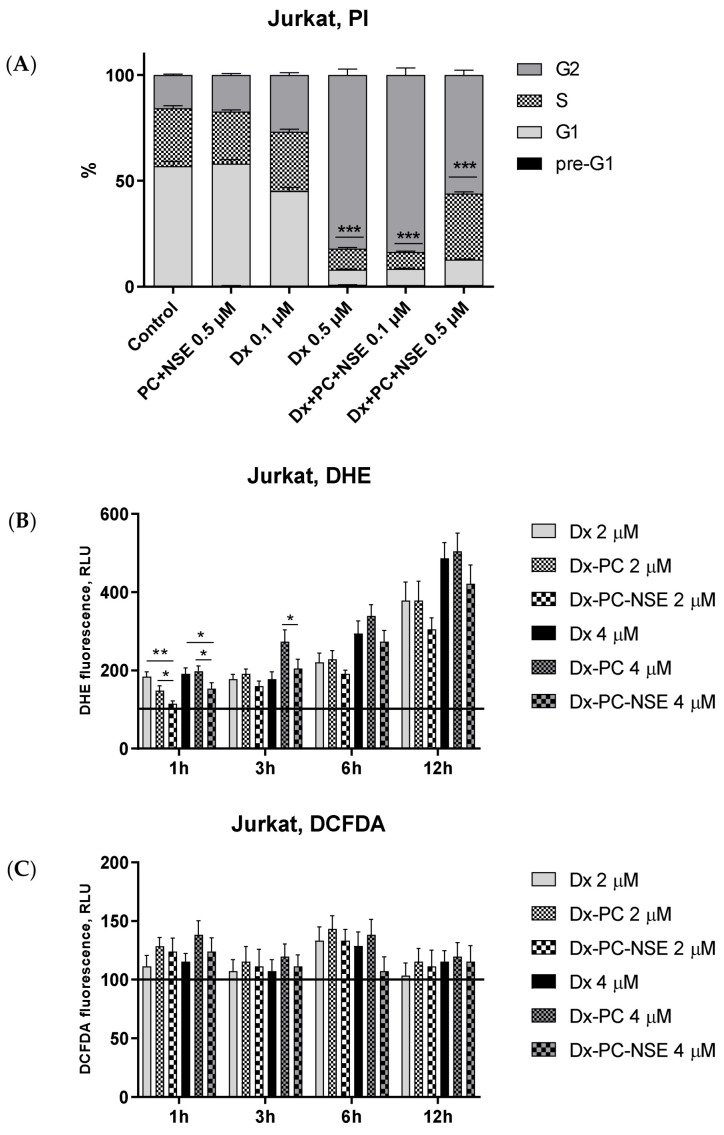
Flow cytometry analysis of impact of co-immobilization of Dx and NSE on PC on cell cycle distribution and ROS induction in Jurkat cells. (**A**) Changes in cell cycle distribution, PI staining, 24 h incubation. (**B**) Time-dependent studies of O_2−_ production, DHE assay, 1,3,6,12 h incubation. (**C**) Time-dependent studies of H_2_O_2_ production, DCFDA assay, 1,3,6,12 h incubation. Data are given relative to the untreated control samples (baseline) and represent the mean ± SD of three independent experiments. Significance to other treatments was calculated by Student’s unpaired *t*-test (* *p* < 0.05; ** *p* < 0.01; *** *p* < 0.001).

**Figure 9 pharmaceutics-15-00835-f009:**
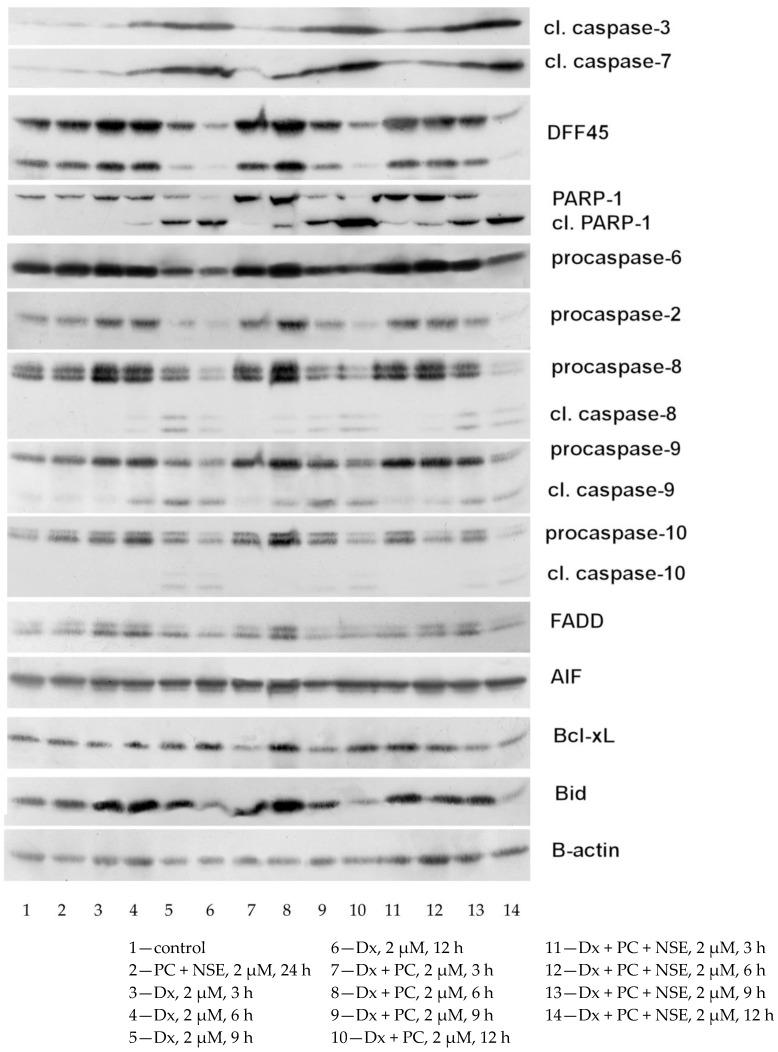
Time-dependent changes in cell death signaling pathways in human Jurkat T-leukemia cells under treatment of various concentrations of Dx, Dx-PC and Dx-PC-NSE.

**Figure 10 pharmaceutics-15-00835-f010:**
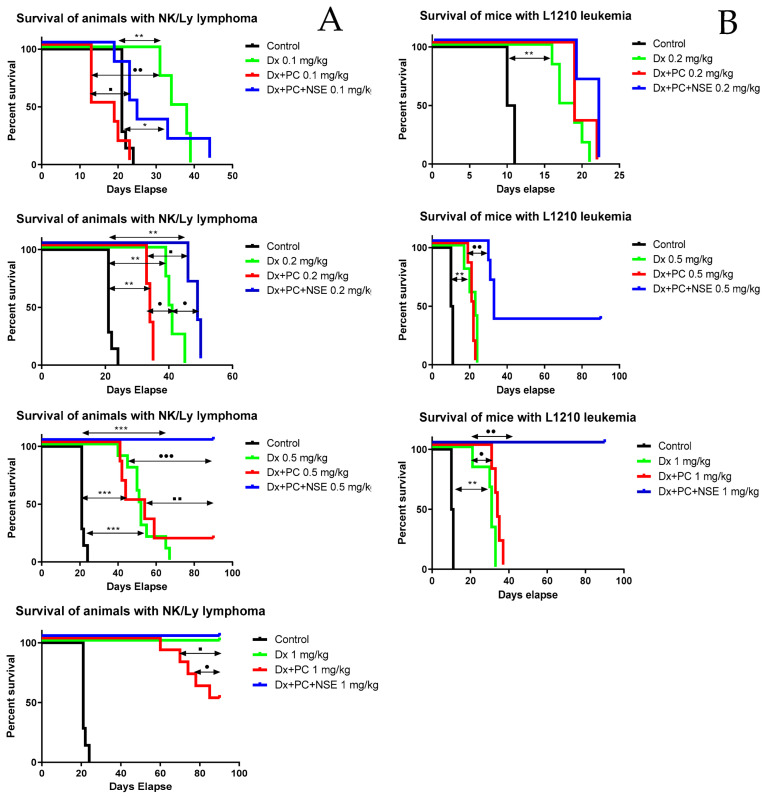
Impact of co-immobilization of Dx and NSE on survival of mice bearing NK/Ly lymphoma (**A**) and L1210 leukemia (**B**), respectively. Significance to other treatments was calculated by Mantel–Cox test (* *p* < 0.05; ** *p* < 0.01; *** *p* < 0.001 relative to control, • *p* < 0.05; •• *p* < 0.01; ••• *p* < 0.001 relative to Dx, ▪ *p* < 0.05; ▪▪ *p* < 0.01 relative to Dx-PC).

**Figure 11 pharmaceutics-15-00835-f011:**
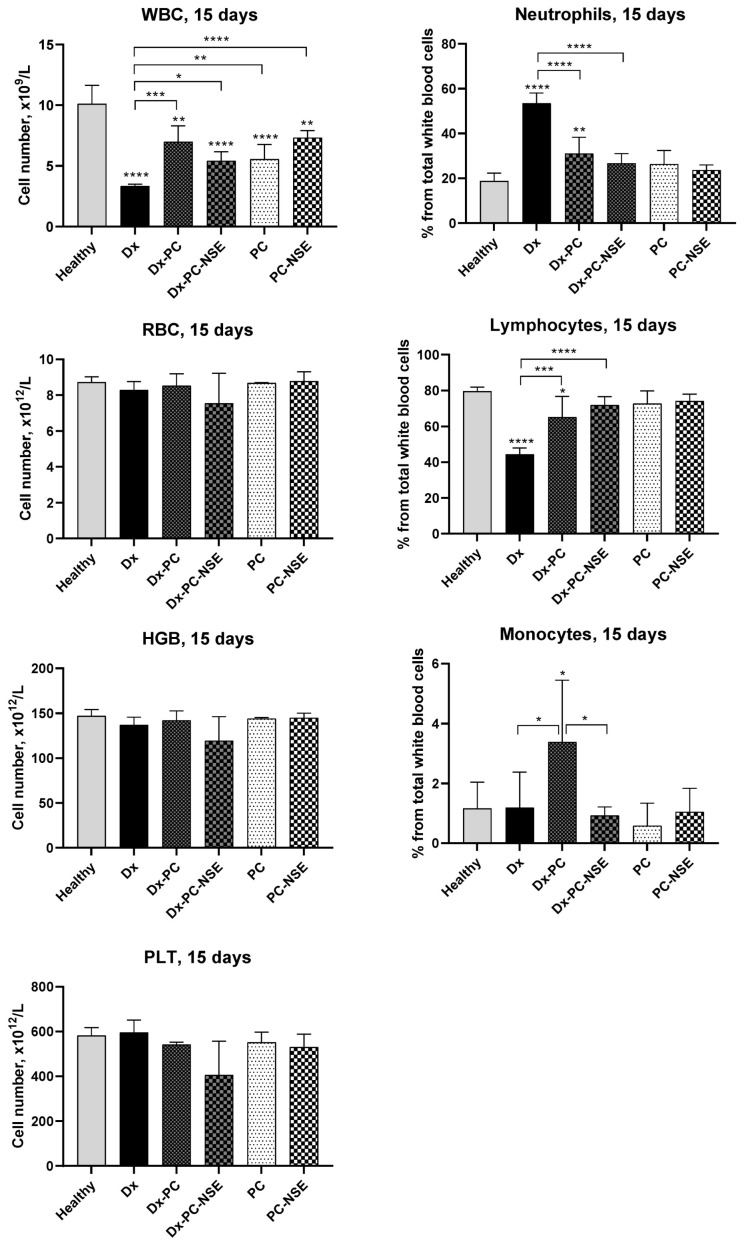
Changes in blood formula in Balb/c mice under treatment with sub-lethal dose of Dx (20 mg/kg), Dx-PC, and Dx-PC-NSE (n = 5). Significance to control and other groups was calculated with one-way ANOVA and Tukey’s multiple comparison test (* *p* ≤ 0.05, ** *p* ≤ 0.01, *** *p* < 0.001, **** *p* < 0.0001).

**Figure 12 pharmaceutics-15-00835-f012:**
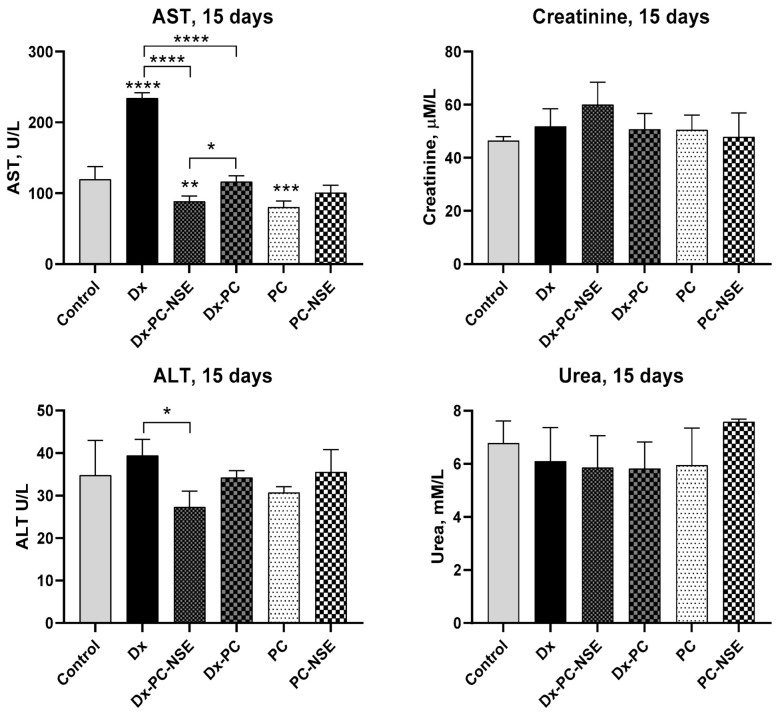
Changes in markers of hepato-and nephrotoxicity in Balb/c mice under treatment with sub-lethal dose of Dx (20 mg/kg), Dx-PC, and Dx-PC-NSE (n = 5). Significance to control and other groups was calculated with one-way ANOVA and Tukey’s multiple comparison test (* *p* ≤ 0.05, ** *p* ≤ 0.01, *** *p* < 0.001, **** *p* < 0.0001).

**Table 1 pharmaceutics-15-00835-t001:** Physico-chemical characteristics of the polymeric carrier poly(VEP-co-GMA)-graft-mPEG and its complex with NSE and Dx.

Sample	Composition of the Water Dispersion mg/mL			DLS Z-Average Hydrodynamic Diameter (nm)	Polydispersity Index (PDI)	Zeta Potential, (mV)
	[PC]	[NSE]	[Dx]			
PC	10	-	-	51 ± 16	0.35	−0.11
PC + NSE	10	0.3	-	420 ± 120	0.25	+4.25
PC + Dx	10	-	0.3	40 ± 9	0.06	+1.57
PC + NSE + Dx	10	0.3	0.3	680 ± 110	0.17	+6.10

**Table 2 pharmaceutics-15-00835-t002:** LC_50_ values of Dx, Dx-PC, and Dx-PC-NSE towards mammalian tumor cell lines in vitro. 24 h incubation, trypan blue assay.

Compound		LC_50_ Values of Compounds for Cell Line, µM (M ± SD)						
	Jurkat	MCF-7	L1210	HeLa	SW1573	SW1573/ 2R160	HL-60	HL-60/adr
Dx	0.52 ± 0.09	0.71 ± 0.04	0.49 ± 0.04	1.03 ± 0.15	1.03 ± 0.06	11.02 ± 0.12	0.17 ± 0.02	37.99 ± 0.56
Dx-PC	0.45 ± 0.13	0.42 ± 0.05	0.32 ± 0.05	0.98 ± 0.18	0.13 ± 0.01	9.32 ± 0.11	0.19 ± 0.03	11.11 ± 0.21
Dx-PC-NSE	0.32 ± 0.07	0.20 ± 0.01	0.02 ± 0.01	0.53 ± 0.09	0.05 ± 0.005	0.85 ± 0.07	0.09 ± 0.01	6.24 ± 0.09

**Table 3 pharmaceutics-15-00835-t003:** Phospholipid profile in heart of mice treated with doxorubicin (7.5 mg/kg, i.p.) and NSE (350 mg/kg, per os).

Phospholipid		Untreated	NSE	Dx	Dx + NSE	NSE, Dx
Phosphatidylcholine	PCh	39.146 ± 0.391	36.895 ± 1.564	43.485 ± 2.403	39.24 ± 1.757	41.119 ± 2.622
Phosphatidylethanolamine	PE	31.246 ± 1.909	33.492 ± 1.659	35.177 ± 2.029	32.676 ± 1.814	34.733 ± 2.91
Diphosphoglycerol	DPG	11.345 ± 0.852	12.227 ± 0.309	6.351 ± 1.08 *@	11.493 ± 0.928 #	11.471 ± 1.186 #
Sphingomyelin	SM	5.163 ± 1.036	6.385 ± 0.643	3.755 ± 0.861 @	7.904 ± 1.208 #	2.978 ± 0.935 @$
Phosphatidylinositol	PI	3.528 ± 0.328	4.551 ± 1.236	6.56 ± 1.133 *	2.01 ± 0.208 *#	3.933 ± 1.007
Phosphatidylserine	PS	2.65 ± 0.486	3.916 ± 0.337	8.977 ± 1.858 *@	4.008 ± 0.83 #	3.064 ± 0.601 #
Lysophosphatidylcholine	LPC	0.924 ± 0.039	1.55 ± 0.351	3.667 ± 0.333 *@	2.343 ± 0.087 *#	2.788 ± 0.693 *

*—*p* < 0.05 compared to control (untreated mice). #—*p* < 0.05 compared to Dx. @—*p* < 0.05 compared to NSE. $—*p* < 0.05 compared to Dx + NSE (simultaneously).

## Data Availability

All data relevant to the publication are included.

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
