# Peer review of "Cannabimimetic N-Stearoylethanolamine as “Double-Edged Sword” in Anticancer Chemotherapy: Proapoptotic Effect on Tumor Cells and Suppression of Tumor Growth versus Its Bio-Protective Actions in Complex with Polymeric Carrier on General Toxicity of Doxorubicin In Vivo"

_pharmaceutics, 2023, doi:10.3390/pharmaceutics15030835_

Round 1

Reviewer 1 Report

In the article titled:” Cannabimimetic N-stearoylethanolamine as “double-edged sword” in anticancer chemotherapy: proapoptotic effect on tumor cells and suppression of tumor growth versus its bio-protective actions in complex with polymeric carrier on general toxicity of doxorubicin in vivo” R. Panchuk and collegues attempted to improve the further treatment of very aggressive and still incurable cancers with synthetic cannabimimetic N-stearoylethanolamine (NSE). As the current treatment of most of leukemias and lymphomas, breast cancers or cervix carcinomas is very limited due to or serious side effects, or insufficient effectiveness there is a constant need to explore new cytotoxic compounds or improve the well known ones. Regarding this issue choosing NSE for further examination seems to a great idea. In the article, authors proved that NSE may both exert cytotoxic effects against certain cancer cells as a single agents and enhance the entry of Doxorubicin into cancer cells and decrease its side effects. It should be highly emphasized that in order to achieve these results, the authors conducted research in a very detailed manner. However, during reading this manuscript I found some linguistic errors that makes the text not easy to follow. I also wonder what made authors to use Doxorubicin Hydrochloride for injection instead to use doxorubicin hydrochloride (in powder), that as other agents is commonly used for the experiments with cell lines.

Author Response

Thanks for your question! Firstly, we consider that using the pharmaceutical form of the Doxorubicin seems to be more adequate at treatment of tumor-bearing animals, than using this agent as a biochemical reagent. Secondly, purchasing of the reagents from the international companies is rather problematic for the Ukrainian scientists, since there are still no representatives of those companies in Ukraine, and the transportation of the reagents is done through a couple of intermediate representatives, usually, in the Western Europe and then, in the Eastern Europe. From our experience, this may lead to a loss of specific activity of some reagents. Consequently, we decided to use the clinically applied preparation of doxorubicin for the current study.   

Reviewer 2 Report

This manuscript report a dose-dependent pro-apoptotic action of synthetic cannabimimetic N-stearoylethanolamine (NSE) on diverse cancer cell lines. The co-immobilization of NSE and doxorubicin on a polymer carrier to form micelles resulted in 2-10-fold increase in anticancer activity. The authors then explored that the reason why doxorubicin-load micelles enhanced the antitumor effect is that the NSE-containing carrier can enhance the accumulation of dox in the tumor cell nucleus. As a delivery system, nanomicelle is relatively mature, and doxorubicin's anti-tumor mechanism is also mature. Therefore, the basement of this design seems not enough innovation, but the experimental design is relatively scientific. Minor revision is suggested for this work, and some detailed comments are followed:

1.     The abbreviation for doxorubicin appears in several forms in this manuscript, including DOX, Dox, and Dx. It is recommended that the authors standardize the abbreviation of doxorubicin.

2.     The authors used DLS to characterize the particle size of nanomicelles, but this was insufficient. It is suggested to use transmission electron microscopy (TEM) or scanning electron microscopy (SEM) to characterize the size and morphology of nanomicelles.

3.     As a drug delivery system, some important studies point to supplementation, such as determination of Dox-loading content (DLC) and In vitro release of DOX from polymer micelles

4.     In part 3.1 of the Results section, the author mention that “A potential reason for this activity of NSE might be its incorporation into the plasma membrane of treated cells that affects membrane structure.” This hypothesis involves changes in the structure of the plasma membrane, and only confocal results (Supp. Fig.1) supported it are clearly insufficient. It is suggested that the authors provide changes in membrane structure imaged via transmission electron microscopy (TEM).

5.     The absence of a title on the vertical axis in Figure 3C will confuse readers. It is advised to include it. Only "%" was indicated on the vertical axis of the analogous experiment result in Figure 7C. It is advised that the author define this proportion in detail. And the order of the figure is marked with A, B, C, but the legend uses (a), (b), (c), it is recommended that the author unify the format.

6.     In part 3.2 of the Result section, the author used different doses of Dox (1 mM and 0.01mM) and different treatment time (24 h and 48 h) to explain whether NSE is capable of modifying the cytotoxic activity of Dox. What I'm confused about is whether the difference is caused by the treatment time or the Dox concentration? The author needs to provide a more thorough and precise description of the experimental findings in the results section.

7.     The WB results for protein cl. caspase-3, cl.caspase-7, procaspase-8, cl.caspase-8, procaspase-10, FADD, and Bid are not suitable for publication, and it is suggested that this experiment be redone.

Author Response

Dear Reviewers!

Thanks a lot for your work on reviewing our manuscript.

We believe that our responses on your remarks and comments will improve our article and make it more scientifically relevant

Remarks

  1. The abbreviation for doxorubicin appears in several forms in this manuscript, including DOX, Dox, and Dx. It is recommended that the authors standardize the abbreviation of doxorubicin.

Response 1

Thanks for this remark, we are sorry for this mistake. It has been corrected.

  1. The authors used DLS to characterize the particle size of nanomicelles, but this was insufficient. It is suggested to use transmission electron microscopy (TEM) or scanning electron microscopy (SEM) to characterize the size and morphology of nanomicelles.

Response 2

Thanks for your advice. We agree with your suggestion and plan to conduct such experiments in near future. However, in order to do that, we will need to prepare a fresh preparation of the particles, that may take more than 20 days allowed by the Editor for the manuscript revision.

  1. As a drug delivery system, some important studies point to supplementation, such as determination of Dox-loading content (DLC) and In vitro release of DOX from polymer micelles

Response 3

Thanks for your recommendation. Results of time-dependent studies of release of Dx from polymer micelles in vitro are presented in Fig.7A and compared to quantitative analysis of Dx entry into Jurkat T-leukemia cells, measured by FACS (Fig. 7B). It was revealed that presence of NSE in structure of polymeric carrier leads to more rapid entry of Dx into cells, which is further accompanied by enhanced release of Dx from PC-NSE at the 75 minute time-point. We thank the Reviewer for the recommendation which helped us to better understand the mode of action of Dx-PC-NSE in vitro.

  1. In part 3.1 of the Results section, the author mention that “A potential reason for this activity of NSE might be its incorporation into the plasma membrane of treated cells that affects membrane structure.” This hypothesis involves changes in the structure of the plasma membrane, and only confocal results (Supp. Fig.1) supported it are clearly insufficient. It is suggested that the authors provide changes in membrane structure imaged via transmission electron microscopy (TEM).

Response 4

Thanks for your recommendation. Unfortunately, due to limited time for revision we were unable to perform the suggested TEM studies. However, we performed extra experiments with NSE at 48h and revealed that despite a significant decrease in the number of Jurkat T-cells under treatment with high doses of NSE, the number of apoptotic (annexin V(+)) cells was actually decreased! These data serve as additional, though indirect clue that NSE leads to inhibition of cell proliferation without cell death induction, presumably by integrating into the cell membrane.

  1. The absence of a title on the vertical axis in Figure 3C will confuse readers. It is advised to include it. Only "%" was indicated on the vertical axis of the analogous experiment result in Figure 7C. It is advised that the author define this proportion in detail. And the order of the figure is marked with A, B, C, but the legend uses (a), (b), (c), it is recommended that the author unify the format.

Response 5

We thank the reviewer for this helpful comment. Corrected (please, see the revised version). Appropriate changes were done to the Figure 3 and Figure 7 (now Fig. 8).

  1. In part 3.2 of the Result section, the author used different doses of Dox (1 mM and 0.01mM) and different treatment time (24 h and 48 h) to explain whether NSE is capable of modifying the cytotoxic activity of Dox. What I'm confused about is whether the difference is caused by the treatment time or the Dox concentration? The author needs to provide a more thorough and precise description of the experimental findings in the results section.

Response 6

Thanks for your advice. In the revised version, we have redone the experiment, using a wider concentration range for NSE (10,20,30,50 µM), Dx (0.01 µM, 0.05 µM, 0.25 µM, 0.50 µM). Moreover, we used an additional assay – annexin V/PI double staining – for better monitoring of cell death induction by combination of various doses of Dx and NSE. Appropriate changes were introduced to Figure 4 and section 3.2 of the article.

  1. The WB results for protein cl. caspase-3, cl.caspase-7, procaspase-8, cl.caspase-8, procaspase-10, FADD, and Bid are not suitable for publication, and it is suggested that this experiment be redone.

 Response 7

Thanks for your advice. Unfortunately, it was technically impossible to prepare all cell lysates with Dx, Dx-PC and Dx-PC-NSE in the time frame given for revision. However, taking into consideration the new data (namely, Fig. 7 on drug entry studies, Supp Fig. 5 on JC-1 staining, indicating on extremely weak involvement of mitochondria in cell death, induced by high doses of Dx, and thus explaining insignificant levels of cl. caspase-8,-9,-10), the low level cleavage of caspases  in the presented Western blots  appear reasonable. However, if the quality of Fig. 9 (before – Fig. 8) is critical to the Reviewer, we propose to remove the Western blot data from the paper.

Reviewer 3 Report

Authors suggested that the activity of NSE for cell death may be its incorporation into the plasma membrane. Authors should show what the type of the cell death by NSE alone (Apoptosis, necroptosis or others) (Fig3).

The order of Fig3 is strange. Fig 3C should be changed to Fig 3B, and Fig 3B should be changed to Fig 3C.

In Fig4A, authors described such effects were observed upon treatment with a low Dx doses (0.01 μM, 48 h), but it is difficult to understand by the Fig. P-value or something is needed to highlight the finding to make it easier to understand.

In Fig 7, it may be interesting if authors add ROS inhibitors to understand the role of ROS in the anti-cancer effect.

The results of Fig 8 are poor especially of cl. Caspase-3 and -7, and PARP-1.  

Authors described that simultaneous increase in level of monocytes, which is considered a marker of prothrombotic activity of Dx (Fig. 10). If so, authors should examine thrombosis marker such as FDP and D-dimer.

Author Response

Dear Reviewers!

Thanks a lot for your work on reviewing our manuscript.

We believe that our responses on your remarks and comments will improve our article and make it more scientifically relevant

Authors suggested that the activity of NSE for cell death may be its incorporation into the plasma membrane. Authors should show what the type of the cell death by NSE alone (Apoptosis, necroptosis or others) (Fig3).

Response

In order to address this interesting remark of the Reviewer, annexin V/PI double staining of Jurkat cells, treated with various concentrations of NSE (10, 20, 30, 50 µM) was performed after 48 h drug exposure. NSE, despite lowering the number of living Jurkat cells at higher concentrations, effectively decreased basal level of apoptosis in these cells. Moreover, NSE in high doses was able to inhibit Dx-induced apoptosis, despite having a little effect on cell number in trypan blue exclusion assay. These data clearly imply a cytostatic and in parallel anti-apoptotic activity of NSE in vitro. Results are presented in the new versions of Fig. 3 and Fig. 4.

The order of Fig3 is strange. Fig 3C should be changed to Fig 3B, and Fig 3B should be changed to Fig 3C.

Response

Thanks for your advice and sorry for the misunderstanding concerning the Figure order.

Corrected as recommended by the Reviewer.

In Fig4A, authors described such effects were observed upon treatment with a low Dx doses (0.01 μM, 48 h), but it is difficult to understand by the Fig. P-value or something is needed to highlight the finding to make it easier to understand.

Response

We thank the reviewer for the helpful advice. In the revised version, we have redone the experiment, using a wider concentration range for NSE (10,20,30,50 µM), Dx (0.01 µM, 0.05 µM, 0.25 µM, 0.50 µM), and used an additional assay – annexin V/PI double staining -  for better monitoring of cell death induction. Appropriate changes were introduced to Figure 4 and section 3.2 of the article.

In Fig 7, it may be interesting if authors add ROS inhibitors to understand the role of ROS in the anti-cancer effect.

Response

Thanks for this interesting comment. In order to reveal the potential source of Dx- and Dx-PC-NSE-induced ROS, we performed in-depth studies of functional status of mitochondria in Jurkat cells, treaded with the same dose of Dx. However, to our surprise, high doses of Dx (2 µM and 4 µM, respectively) led only to a weak depolarization of mitochondria only at the 9 h time-point, which was further decreased at 12h exposure despite massive cell death, as revealed by trypan blue exclusion assay (see Supp. Fig. 5). We assume that at high concentrations Dx leads to massive DNA intercalation instead of redox-cycling, which would explain the weak involvement of mitochondria. Thus, antioxidants would not be expected to have a strong effect in these experiments.

The results of Fig 8 are poor especially of cl. Caspase-3 and -7, and PARP-1. 

Response

We agree with Reviewer, but due to lack of time, given to us by Editor, it was technically impossible to prepare again the entire set of cell samples for all time-points for Western-blot analysis. However, with new Fig. 7 on drug entry studies and Supp Fig. 5 on mitochondrial status the actual value of current Western blots is diminished, thus, if the quality of Fig. 9 (before – Fig. 8) is critical to the Reviewer, we ask the Editor to delete it from the manuscript. In future studies, we will repeat this experiment which requires conducting several electrophoresis procedures and membrane re-probing.

Authors described that simultaneous increase in level of monocytes, which is considered a marker of prothrombotic activity of Dx (Fig. 10). If so, authors should examine thrombosis marker such as FDP and D-dimer.

Response

That is an interesting suggestion, however, we are unable to perform the respective in vivo experiment during the short revision time proposed by the Editor.

Round 2

Reviewer 3 Report

The manuscript was modified, and I have no more comment.